# CLUSTERING BY DENOISING: LATENT PLUG-AND-PLAY DIFFUSION FOR SINGLE-CELL DATA

**Dominik Meier**
Cornell Tech
dm954@cornell.edu

**Shixing Yu**
Cornell Tech
sy774@cornell.edu

**Sagnik Nandy**
Ohio State University
nandy.15@osu.edu

**Promit Ghosal**
University of Chicago
promit@uchicago.edu

**Kyra Gan**
Cornell Tech
kyragan@cornell.edu

## ABSTRACT

Single-cell RNA sequencing (scRNA-seq) enables the study of cellular heterogeneity. Yet, clustering accuracy, and with it downstream analyses based on cell labels, remain challenging due to measurement noise and biological variability. In standard latent spaces (e.g., obtained through PCA), data from different cell types can be projected close together, making accurate clustering difficult. We introduce a *latent plug-and-play diffusion* framework that *separates the observation and denoising space*. This separation is operationalized through a novel Gibbs sampling procedure: the learned diffusion prior is applied in a low-dimensional latent space to perform denoising, while to steer this process, noise is *reintroduced into the original high-dimensional observation space*. This unique "input-space steering" ensures the denoising trajectory remains faithful to the original data structure. Our approach offers three key advantages: (1) *adaptive noise handling* via a tunable balance between prior and observed data; (2) *uncertainty quantification* through principled uncertainty estimates for downstream analysis; and (3) *generalizable denoising* by leveraging clean reference data to denoise noisier datasets, and via averaging, *improve quality beyond the training set*. We evaluate robustness on both synthetic and real single-cell genomics data. Our method improves clustering accuracy on synthetic data across varied noise levels and dataset shifts. On real-world single-cell data, our method demonstrates improved biological coherence in the resulting cell clusters, with cluster boundaries that better align with known cell type markers and developmental trajectories.

## 1 INTRODUCTION

Single-cell RNA sequencing (scRNA-seq) has revolutionized biomedical research by enabling high-resolution profiling of cellular heterogeneity (Park et al., 2020; Miragaia et al., 2019), with large-scale initiatives like the Human Cell Atlas providing foundational references for *cell type annotation* (Regev et al., 2017; Lindeboom et al., 2021; Elmentaite et al., 2022; Stuart et al., 2019; Lopez et al., 2018a). However, since cells are sequenced *without predefined labels*, accurate cell type identification must be derived entirely from *unsupervised analysis* of noisy high-dimensional data. The canonical approach involves reducing dimensionality (e.g., via PCA) followed by clustering and manual annotation based on marker genes—an iterative and subjective process (Kiselev et al., 2019; Stuart et al., 2019). However, this entire paradigm is often undermined by the inherent noise in scRNA-seq data, which arises from technical artifacts like varying capture efficiency (Kharchenko et al., 2014) and biological stochasticity (Wagner et al., 2016), which are amplified by standard clustering algorithms and lead to unreliable labels.

We frame single-cell denoising as an inverse problem: recovering clean gene expression from noisy measurements without imposing strong generative assumptions. Our method leverages the high-quality signal from reference datasets (e.g., SMART-seq2, Picelli et al. 2013; 2014) to learn a powerful denoising prior, which is then applied to enhance noisier target data (e.g., droplet-based scRNA-seq, Klein et al. 2015; Macosko et al. 2015) for improved clustering. We operationalize

this under the *plug-and-play* (PnP) paradigm (Venkatakrishnan et al., 2013a; Zhang et al., 2021; Chan et al., 2016; Ryu et al., 2019), which integrates powerful denoising priors with measurement models. Mainstream PnP diffusion frameworks (Zhu et al., 2023; Go et al., 2023; Wu et al., 2024; Coeurdoux et al., 2024; Xu & Chi, 2024) enable this through combining likelihoods via iterative refinement (e.g., Gibbs sampling), where each denoising step is followed by controlled noise reintroduction to enforce data consistency. This principled approach enables us to denoise beyond test data quality by transferring patterns from high-signal reference data to noisier technologies, even under mild distribution shifts between the reference and target distribution. However, directly applying image-based PnP frameworks to single-cell data is challenging. Unlike images where pixel noise is largely independent, gene expression data exhibits intrinsic low-rank structure and complex correlations. Moreover, denoising must preserve relational structure between cells for accurate clustering and annotation. Standard dimensionality reduction (e.g., PCA) can collapse distinct cell types, making it impossible to guide denoising accurately.

To address these unique challenges, we introduce a *latent plug-and-play diffusion framework* that tailors the PnP philosophy to single-cell biology. Unlike prior PnP methods that rely on generic or hand-designed priors, our approach is tailored to biological data and operates through a *modified two-step procedure*, designed to overcome limitations of prior single-cell denoising methods. *First*, a diffusion model is trained in a low-dimensional latent space (analogous to PCA) to capture the core biological manifold of *a high-quality reference dataset*. Unlike PCA, this diffusion process learns score functions directly from the data, enabling recovery of complex structures in the latent distribution of cell types. This approach inherits the robustness to prior misspecification and scalability to high latent dimensions characteristic of diffusion models (Xu & Chi, 2024). *Second*, during inference on the *noisy dataset*, we employ a *Gibbs sampling procedure that reintroduces noise into the original high-dimensional input space*. This critical step directly addresses the *latent-space collapsing issue* inherent in methods like PCA (Burges et al., 2010), where distinct biological states are projected too close together, losing information essential for precise denoising. By operating in the original high-dimensional space, where full geometric relationships are preserved, our method steers denoising toward biologically meaningful structures obscured in compressed representations.

Our framework diverges from existing Bayesian approaches for single-cell analysis by removing their need for restrictive generative modeling. While variational autoencoders (VAEs) (Lopez et al., 2018a; Gayoso et al., 2021; Grønbech et al., 2020) are difficult to train and rely on strong likelihood assumptions, more recent *approximate message passing* methods with empirical Bayes denoisers (Zhong et al., 2022; Nandy & Ma, 2024) still require parametric noise modeling, operate purely in the latent space, and scale poorly to high-dimensional latent spaces. In contrast, our approach requires no explicit generative model or pre-processing for noise structure, instead learning it directly from data.

By combining the adaptability of likelihood-free diffusion with the structure-aware refinement of Gibbs-based input-space guidance, we enhance cluster separation in the denoised low-dimensional embeddings of the target data, even when the biological signal is weak or the target distribution moderately differs from the reference.

Our diffusion based denoising framework offers three key advantages over existing single-cell denoising methods, and therefore provides a principled, robust, and reproducible pathway for diverse downstream tasks like automated cell type annotation using the denoised embeddings.

- **Adaptive noise handling through tunable interpolation:** We introduce a parameter $\rho$ that dynamically balances data-driven information and prior knowledge during denoising. This allows optimal adaptation to varying noise levels and dataset qualities—preserving data-specific signals when test and training distributions align, while leveraging prior knowledge to stabilize highly noisy inputs. This capability is absent in conventional clustering and imputation methods.
- **Uncertainty quantification:** Unlike standard clustering or VAE-based pipelines, our approach provides confidence sets for cell-type predictions, enabling quantitative assessment of annotation reliability—critical for downstream analysis and clinical applications.
- **Generalizable denoising:** By training on high-quality reference data, our model learns a robust biological manifold that can denoise even low-quality target datasets, effectively addressing real-world scenarios where data from different labs exhibit substantial quality variations. Further, our averaging-based approach enables denoising beyond the immediate training distribution, enhancing applicability across diverse experimental conditions.

Our experimental results demonstrate consistent performance under various mis-specifications of the data-generating process in synthetic settings (Section 4). In real-world single-cell experiments, our method shows strong potential for leveraging high-signal training data to improve denoising in low-signal datasets and to denoise beyond the training distribution by averaging (Section 5). The source code of DICE is publicly available.[1]

## 1.1 ADDITIONAL RELATED WORK

**Diffusion in Single-Cell Data**   Diffusion models provide flexible, trainable priors that accommodate complex noise structures (Sohl-Dickstein et al., 2015; Ho et al., 2020; Song et al., 2021b;a). While initial single-cell applications have focused on transcriptome generation and data imputation (Luo et al., 2024; Wang et al., 2025; 2024; Zhang et al., 2025), their potential for learning denoised low-dimensional embeddings, critical for reference atlas construction and robust label transfer, remains largely unexplored beyond restrictive generative modelling assumptions. Our work investigates this gap, using unsupervised clustering and subsequent biological validation as the evaluation framework.

**Single-Cell Preprocessing**   Analyzing single-cell data is challenging because raw UMI counts vary across labs, technologies, and populations (Shaham et al., 2017). Several pipelines have been proposed aim to mitigate these effects, most prominently the ubiquitous Seurat workflow (Stuart et al., 2019) applying quality control, and normalization. In datasets with pronounced batch effects, such batch-induced variations are typically corrected using techniques like Harmony (Korsunsky et al., 2019), and the batch corrected expressions are used to produce a linear embedding for downstream tasks such as clustering or label transfer. An alternative direction incorporates these steps into deep generative models that operate directly on raw counts, such as scVI (Lopez et al., 2018b), which learns latent representations end-to-end.

Instead of creating embeddings for all cells, a complementary line of work aggregates similar cells into metacells (e.g., MetaQ (Li et al., 2025)), which are then used for downstream analyses. In contrast, our method produces embeddings for all individual cells, rather than summarizing them.

## 2 SINGLE-CELL ATLAS CONSTRUCTION VIA POSTERIOR SAMPLING

This work considers two single-cell RNA-seq datasets: (1) a reference dataset $\mathcal{D}^{(r)} = \left\{ X_1^{(r)}, \ldots, X_m^{(r)} \right\}$ used to train a diffusion prior that captures the underlying biological manifold, and (2) a target dataset $\mathcal{D}^{(t)} = \left\{ X_1^{(t)}, \ldots, X_n^{(t)} \right\}$ for denoising. Our framework is designed for single-cell RNA-seq datasets after standard preprocessing steps, such as quality control to filter low-quality cells, library-size normalization, and a $\log 1p$ transform which renders the residual non-biological variation approximately Gaussian. Within this regime, our method does not rely on stronger assumptions about upstream preprocessing choices. The goal is to construct a *single-cell atlas*—a reference map of cellular states— learned via denoised low-dimensional embeddings that capture underlying biological structure while removing technical variation arising from non-biological sources.

**Data Generating Process**   Both datasets are modeled using a low-rank factor model (Peng et al., 2021; Weine et al., 2024; Zhong et al., 2022; Nandy & Ma, 2024; Argelaguet et al., 2020). Let $X_i \in \mathbb{R}^d$ represent the gene expression profile for cell $i$:

$$X_i = VU_i + \varepsilon_i, \quad \forall i, \tag{1}$$

where $V \in \mathbb{R}^{d \times k}$ is a factor loading matrix that spans the transcriptional space; $U_i \in \mathbb{R}^k \sim P_{\text{prior}}$ encodes the low-dimensional *biological cell signal* drawn i.i.d. from an *unknown* distribution $P_{\text{prior}}$; and $\varepsilon_i$ is an independent noise vector, reflecting stochastic variation in the measurement process or biological stochasticity.

---

[1] https://github.com/dommeier/dice

Our method is built on a generative model that projects both reference and target datasets into a shared latent space. This is formalized as:

$$X^{(r)} = U^{(r)} \left(V^{(r)}\right)^\top + \varepsilon^{(r)}, \quad U^{(r)} \in \mathbb{R}^{m \times k}, \quad \varepsilon^{(r)} \in \mathbb{R}^{m \times d}; \tag{2}$$

$$X^{(t)} = U^{(t)} \left(V^{(r)}\right)^\top + \varepsilon^{(t)}, \quad U^{(t)} \in \mathbb{R}^{n \times k}, \quad \varepsilon^{(t)} \in \mathbb{R}^{n \times d}. \tag{3}$$

The critical modeling choice is the shared factor loading matrix $V$, learned from the reference dataset. This allows the target data to be projected into the same latent space as the reference, enabling knowledge transfer. This approach explicitly accommodates different measurement technologies through distinct noise models for each dataset ($\varepsilon^{(t)} \nsim \varepsilon^{(r)}$), *with the expectation that our method performs best when the reference dataset $\mathcal{D}^{(r)}$ has comparable or lower noise levels than the target dataset $\mathcal{D}^{(t)}$.*

Given this shared structure, we drop superscripts on $V$ entirely. We retain the $(r)$ and $(t)$ superscripts only for dataset-specific terms: $\mathcal{D}^{(r)}, \mathcal{D}^{(t)}, X^{(r)}, X^{(t)}, U^{(r)}, U^{(t)}, \varepsilon^{(r)},$ and $\varepsilon^{(t)}$. To further simplify notation, we henceforth drop the superscripts with the convention that all quantities refer to the target dataset $\mathcal{D}^{(t)}$ unless explicitly marked with $(r)$ for the reference dataset $\mathcal{D}^{(r)}$.

**Posterior Sampling for Denoised Embeddings**  Our goal is to construct a *denoised atlas* of $\mathcal{D}^{(t)}$ by computing posterior embeddings $\mathbb{E}[U_i \mid X_i]$, framed as sampling from the posterior:

$$\pi(U \mid X) \; \propto \; f\left(X - UV^\top \mid U\right) \; P_{\text{prior}}(U),$$

where $f(\cdot)$ denotes the likelihood associated with the observation model, and $P_{\text{prior}}$ represents the population prior learned via diffusion on $\mathcal{D}^{(r)}$. The main challenge in sampling arises from the joint influence of likelihood and prior. Efficiently sampling from a posterior that couples these two components is non-trivial, as it requires balancing the local reconstruction constraints encoded by the likelihood with the global manifold structure enforced by the prior. Traditional approaches address this difficulty by imposing restrictive assumptions: conjugate Gaussian priors with Gaussian likelihoods (Gelman et al., 2013) enable tractable inference but fail to capture complex biological distributions, while Metropolis-adjusted Langevin algorithms (Roberts & Rosenthal, 1998; Durmus et al., 2018) handle non-Gaussian likelihoods but struggle with implicitly defined diffusion priors.

**Remark 2.1** (Design Principle: Reference-Guided Denoising)**.** *A core design principle of this method is to leverage the high-quality reference dataset to learn a biologically meaningful prior, $P_{prior}(U)$, which captures the latent manifold of plausible cell states. This prior is not used to force the target dataset into an identical configuration, but to provide a structural guide for denoising. The shared projection matrix $V$ enables this by projecting both datasets into a common latent space, allowing the diffusion prior learned from $\mathcal{D}^{(r)}$ to provide a structural guide for denoising the target data. By projecting the noisier target data into this well-defined space, our diffusion prior can effectively separate biological signal from dataset-specific noise while accommodating different measurement technologies through distinct noise models.*

**PnP Framework with Auxiliary Variables**  By leveraging the PnP diffusion framework, we overcome this challenge using a *split Gibbs sampling* approach (Vono et al., 2019; Xu & Chi, 2024) that introduces auxiliary variables $Z_i$ to decouple the likelihood from the diffusion prior. This is achieved by first replacing $U_i$ with $Z_i$ in the likelihood generating stage, and then enforce consistency between $U_i$ and $Z_i$ through a Gaussian penalty, which leads to following augmented joint distribution:

$$P_\rho(X_i, U_i, Z_i) \; \propto \; \exp\left(-\log f(X_i - VZ_i) - \frac{1}{2\rho^2}\|U_i - Z_i\|_2^2 - \log P_{\text{prior}}(U_i)\right), \tag{4}$$

where $\rho$ controls the alignment strength. Smaller $\rho$ enforces tighter coupling between $U_i$ and its auxiliary counterpart $Z_i$. This leads to a Gibbs sampler alternating between two conditional updates:

**Likelihood Step:**  $P_\rho(Z_i \mid X_i, U_i) \; \propto \; \exp\left(-\log f(X_i - VZ_i) - \frac{1}{2\rho^2}\|U_i - Z_i\|_2^2\right),$  (5)

**Prior Step:**  $P_\rho(U_i \mid Z_i) \; \propto \; \exp\left(-\frac{1}{2\rho^2}\|U_i - Z_i\|_2^2 - \log P_{\text{prior}}(U_i)\right).$  (6)

Although resembling Gibbs sampling, the alignment penalty is *artificially introduced* rather than arising from standard conjugacy. This modification allows us to plug in a diffusion prior at inference time, thereby enabling efficient posterior sampling even when the prior is only implicitly specified.

**Evaluating Atlas Quality**   A key challenge in single-cell atlas construction is the lack of direct evaluation metrics due to high-dimensional noise and the curse of dimensionality (Kiselev et al., 2019). We employ a multi-faceted evaluation strategy combining quantitative clustering metrics and qualitative visual assessment:

1. *Visual Assessment:* We use dimensionality reduction techniques, particularly UMAP (McInnes et al., 2018), to visually examine the quality of denoised embeddings. Well-separated, biologically meaningful structures in 2D visualizations validate the method's performance.

2. *Unsupervised Clustering with Post-Hoc Evaluation:* Denoised embeddings $\{\widehat{U}_i\}_{i=1}^n$ are clustered without using label information to obtain $K$ clusters. The quality of these clusters is evaluated by comparing them to the known cell type labels $L_i \in 1, \ldots, C$ using the metrics *adjusted rand index* (ARI) (Hubert & Arabie, 1985), *average silhouette score* (Rousseeuw, 1987), *normalized mutual information* (NMI) (Strehl & Ghosh, 2002), and *cell type locally invariant Simpson's index* (cLISI) Korsunsky et al. (2019),[2] which quantify the agreement between the data-driven clusters and biological annotations.

Our framework is fully *unsupervised*, relying only on expression data.[3] We benchmark performance on synthetic data with known ground truth and on real datasets, noting that real-world labels may be imperfect (and thus annotation accuracy provides an indirect measure of atlas quality).

## 3   METHOD

We introduce a *latent plug-and-play diffusion scheme* for denoising query cell embeddings under the guidance of a diffusion model trained on a large corpus of single-cell gene expression data. Leveraging the PnP framework, we recover low-dimensional embeddings $U$ from noisy gene expression profiles $X^{(r)}$ by *decoupling data fidelity and prior structure*: the query cell's expression profile anchors the denoising process to its unique features, while the pretrained diffusion model contributes global information about the structure of the cell population. Our pipeline consists of two stages:

1. **Training stage:** jointly estimate the factor loading matrix $V$ and train a diffusion model to learn the prior distribution $P_{\text{prior}}$ over embeddings on $\mathcal{D}^{(r)}$.

2. **Inference stage:** given query expressions $X_q^{(t)}$, perform posterior sampling using a split Gibbs scheme, alternating between likelihood-informed updates and diffusion-guided updates.

An illustration of the method can be found in Figure A.1. This approach preserves the flexibility of diffusion priors while maintaining tractable posterior inference, providing a scalable and uncertainty-aware framework for single-cell atlas construction.

**Diffusion Training**   We train the diffusion model on the reference dataset $\mathcal{D}^{(r)}$. Consider the best rank-$k$ approximation of the reference data produced by singular value decomposition: $X^{(r)} \approx \widehat{W}\widehat{\Sigma}\widehat{V}^\top$, where $\widehat{W} \in \mathbb{R}^{n \times k}$ and $\widehat{V} \in \mathbb{R}^{d \times k}$ are unitary matrices containing the top $k$ left and right singular vectors, respectively, and $\widehat{\Sigma} \in \mathbb{R}^{k \times k}$ is the diagonal matrix of the top $k$ singular values. This decomposition yields our loading matrix estimate in Eq. (1), $\widehat{V}$. We then compute the transformed observations $\widehat{U}_i = \widehat{V}^\top X_i^{(r)}$ for $i = 1, \ldots, m$, which accurately approximates the latent embeddings $U_i$ under a wide range of noise models. These estimated embeddings $\{\widehat{U}_i\}_{i=1}^m$ therefore provide approximate training samples from the prior distribution $P_{\text{prior}}$ and are used to train the diffusion model.

To learn this prior, we adopt the standard forward-diffusion framework (Sohl-Dickstein et al., 2015; Ho et al., 2020), which we detail in Appendix A.

**Denoising with** DICE   We now introduce DICE (Diffusion Induced Cell Embeddings, Algorithm 1), our split Gibbs sampling procedure for denoising a query cell $X_q$ and estimating its la-

---

[2]For detailed description of these metrics, see Appendix D.

[3]While labels are not used in denoising, we explore label-augmented diffusion training in Section 5.

---

**Algorithm 1** DICE: Diffusion Induced Cell Embeddings

---

1: **Input:** query cell $X_q$; trained diffusion model $\hat{\varepsilon}_{\theta_t}(\cdot)$; number of iterations $T$; annealing schedule $\{\rho_s\}_{s=1}^T$; estimated factor loading matrix $\widehat{V}$
2: **Initialize**: $U_q^{(0)} \leftarrow \widehat{V}^\top X_q$
3: **for** $s = 0$ to $T - 1$ **do**
4:     **Likelihood alignment:** sample
$$Z_q^{(s)} \mid U_q^{(s)} \ \propto \ \exp\Big(-\tfrac{1}{2\rho_s^2}\|Z_q - U_q^{(s)}\|_2^2 - \log f(X_q - \widehat{V}Z_q)\Big).$$
5:     **Prior alignment:** Using the reverse diffusion update in Eq. (7), sample
$$U_q^{(s+1)} \mid Z_q^{(s)} \ \propto \ \exp\Big(-\tfrac{1}{2\rho_s^2}\|Z_q^{(s)} - U_q\|_2^2 - \log P_{\text{prior}}(U_q)\Big).$$
6: **end for**
7: **return** $U_q^{(T)}$ as the denoised embedding of the query cell.

---

tent embedding $U_q$. Given an annealing schedule $\{\rho_s : s = 1, \ldots, T\}$, the augmented distribution Eq. (4) decomposes posterior sampling into two iterative steps:

1. **Likelihood alignment** (Line 4) is implemented using either a general proximal scheme (Xu & Chi, 2024) or a closed-form Gaussian update when $f$ is Gaussian (Proposition 3.1). Unlike the prior alignment step in Eq. (5), Line 4 operates in the original $d$-dimensional data space, reintroducing noise through the likelihood function $\log f(X_q - \widehat{V}Z_q)$.

2. **Prior alignment** (Line 5) is implemented using the trained diffusion model via the reverse update

$$x_{t-1} \ = \ \frac{1}{\sqrt{\alpha_t}}\left(x_t - \frac{1 - \alpha_t}{\sqrt{1 - \bar{\alpha}_t}}\,\hat{\varepsilon}_\theta(x_t)\right) + \sqrt{1 - \alpha_t}\,z_t, \qquad z_t \sim \mathcal{N}_k(0, I_k), \tag{7}$$

   run for $t = t_0, t_0 - 1, \ldots, 1$ with initialization $x_{t_0} = \sqrt{\bar{\alpha}_{t_0}}\,U_q^{(s)}$. The chain length is chosen so that $\bar{\alpha}_{t_0} \approx (1 + \rho_s^2)^{-1}$.

We reduce posterior variability by generating multiple samples and averaging, enabling denoising beyond reference data quality (Figure 1 (first column) and Figure 3). The parameter $\rho_s$ controls the relative weight of prior and likelihood: a larger $\rho_s$ emphasizes population-level structure and is suitable for noisy queries, while a smaller $\rho_s$ emphasizes fidelity to the observed expression profile.

**Likelihood Alignment under Gaussian Noise** Although our framework accommodates general likelihoods, in practice, single-cell data are often $\log 1p$-transformed and modeled with Gaussian noise (Zhong et al., 2022; Argelaguet et al., 2020). In this case, Proposition 3.1 (proof in Appendix B) establishes that the likelihood update step admits a closed-form update:

**Proposition 3.1.** *Assume $f$ is the standard multivariate Gaussian density in $d$ dimensions. Following Gaussian conjugacy, for all $s = 0, \ldots, T - 1$, the likelihood update step (Line 4) admits the following update:*
$$Z_q^{(s)} \sim \mathcal{N}_k\Big(\Lambda\Big(\widehat{V}^\top X_q + \tfrac{1}{\rho_s^2}U_q^{(s)}\Big), \Lambda\Big), \qquad \Lambda = \Big(\widehat{V}^\top \widehat{V} + \tfrac{1}{\rho_s^2}I_k\Big)^{-1}.$$

**Remark 3.2.** *The same denoising scheme can be applied to the training data themselves, yielding refined embeddings that serve as a reference atlas. Notably, atlas construction via DICE does not rely on restrictive parametric assumptions for either the likelihood or the prior, enabling the method to capture rich and complex population structures as found for example in single-cell data.*

**Confidence Sets** We quantify uncertainty in the embedding of a query cell $X_q$ by applying DICE multiple times and examining the spread of the resulting denoised embeddings.

## 4 EVALUATION ON SYNTHETIC DATA

**Setup** To evaluate how well DICE recovers clean latent structure from noisy expression profiles, we design a controlled setting that mimics two *pure* cell populations with known labels. In latent dimension $k = 15$, the training prior is a balanced Gaussian mixture $P_{\text{prior}} = \frac{1}{2}\mathcal{N}_{15}\big(0_{15}, 1.5\,I_{15}\big) + \frac{1}{2}\mathcal{N}_{15}\big(1_{15}, 1.3\,I_{15}\big)$, with each component corresponding to one cell type. We sample a loading

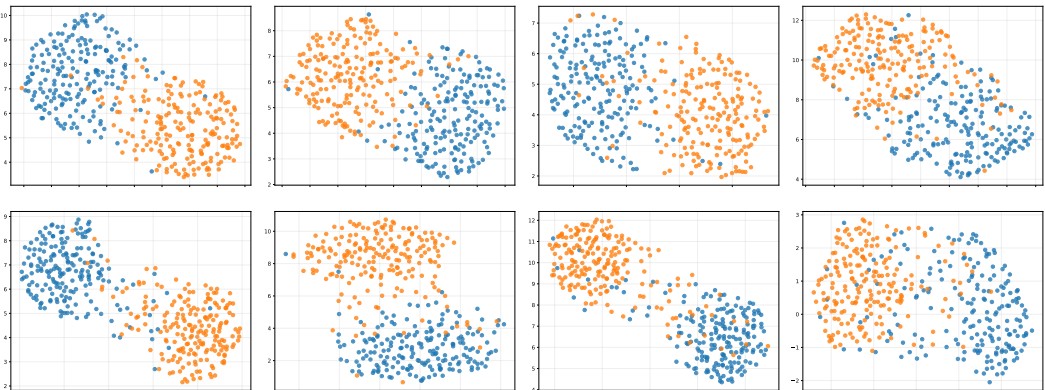

Figure 1: UMAP visualizations of the 400 test cells for each of the four configurations. Top row: PCA projections; bottom row: DICE-denoised embeddings. Columns (left to right) correspond to Setups 1–4. Throughout, we refer to the mixture component centered at $1_{15}$ with scale $1.3\,I_{15}$ as **Cluster 1**, and to the complementary component as **Cluster 2** (Gaussian $\mathcal{N}_{15}(0_{15},\,1.5\,I_{15})$ where applicable; heavy-tailed $\mathtt{t}_{\nu=4}(1.3\,I_{15})$ in the latent–prior–shift configuration). Points are colored blue for **Cluster 1** and orange for **Cluster 2**.

matrix $V \in \mathbb{R}^{2000 \times 15}$ with entries i.i.d. $\mathcal{N}(0,1)$. For each cell $i$, we draw a latent $U_i^{(r)} \sim P_{\text{prior}}$ and measurement noise $\varepsilon_i^{(r)} \sim \mathcal{N}_{2000}(0, I_{2000})$, and generate a synthetic expression profile $X_i^{(r)} = VU_i^{(r)} + \varepsilon_i^{(r)} \in \mathbb{R}^{2000}$, $i = 1, \ldots, 1600$. This yields a reference dataset $\mathcal{D}^{(r)}$ of size $m = 1600$ in $d = 2000$ observed dimensions, analogous to a high-signal scRNA-seq experiment with two subpopulations present in equal proportion.

**Test configurations** We examine three train-test shifts covering practical single-cell scenarios, with $|\mathcal{D}^{(t)}| = 400$, fixing $P_{\text{prior}}$ and $V$ as in Section 2 unless noted otherwise:

1. **Setup 1 (matched train–test distribution).** We generate $\mathcal{D}^{(t)}$ from the same generative distribution as $\mathcal{D}^{(r)}$ to isolate posterior denoising effects. This evaluates whether DICE improves population separation in latent space versus PCA, mirroring standard atlas workflows where query cells are mapped via denoised embeddings (Nandy & Ma, 2024).

2. **Setup 2 (signal-strength shift).** We increase the noise on $\mathcal{D}^{(t)}$, with $\varepsilon_i^{(t)} \sim \mathcal{N}_{2000}(0, 10\,I_{2000})$. This mirrors single-cell scenarios with lower read depth/fewer UMIs or noisier platforms (e.g., shallow sequencing) that reduce *signal-to-noise ratio* (SNR) at test time. This tests whether DICE, trained on high-quality data, denoises low-SNR profiles better than PCA.

3. **Setup 3 (noise-model shift).** We generate $\mathcal{D}^{(t)}$ with a heavy-tailed noise, $\varepsilon_i^{(t)} \sim \mathtt{t}_{\nu=4}(I_{2000})$ (multivariate $\mathtt{t}$ with 4 degrees of freedom and scale $I_{2000}$), while keeping a Gaussian likelihood during denoising. This tests DICE's robustness to likelihood mis-specification from heavy-tailed residuals (due to outliers, doublets, or over-dispersion in single-cell applications) against PCA.

4. **Setup 4 (latent-prior shift).** We change the test latent distribution to a heavy-tailed mixture: $U_i^{(t)} \overset{\text{i.i.d.}}{\sim} \frac{1}{2}\mathcal{N}_{15}(1_{15}, 1.5\,I_{15}) + \frac{1}{2}\mathtt{t}_{\nu=4}(1.3\,I_{15})$, with increased noise $\varepsilon_i^{(t)} \sim \mathcal{N}_{2000}(0, 10\,I_{2000})$. This tests robustness to prior mis-specification when deploying on novel, heterogeneous subpopulations (e.g., new developmental states), where we expect DICE to recover mixture separation better than PCA despite heavier tails and lower SNR.

**Training and denoising workflow** We train a diffusion model on 15-dimensional latent representations obtained via PCA from the training set $\mathcal{D}^{(r)}$ for 2,000 epochs (training details in Appendix C). The resulting model serves as the learned prior $P_{\text{prior}}$ across all four test configurations.

At test time, we denoise the PCA projections of each configuration by running Algorithm 1 (DICE) for $T = 200$ Gibbs iterations with a constant annealing level $\rho_t = 20$. We repeat this procedure with 10 independent random iterations and report the mean of the resulting embeddings as a Monte Carlo estimate of $\mathbb{E}[U \mid X]$ for each test cell.

| Setup | Silhouette | | cLISI | |
|---|---|---|---|---|
| | PCA | DICE | PCA | DICE |
| 1 | 0.25 | **0.37** | 1.27 | **1.17** |
| 2 | 0.24 | **0.36** | 1.27 | **1.17** |
| 3 | 0.22 | **0.34** | 1.32 | **1.18** |
| 4 | 0.22 | **0.28** | 1.35 | **1.27** |

Table 1: Comparsion of average silhouette scores (higher is better) and cLISI scores (lower is better) calculated on the ground-truth cluster labels across four synthetic setups.

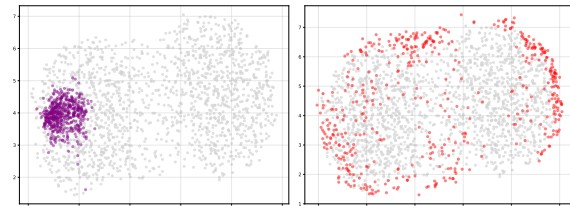

Figure 2: UMAP visualizations of 500 runs of DICE on the same input point in Setup 1 and $\rho = 0.1$. Training data are shown in grey. Left: center of Cluster 2; Right: midpoint between Clusters 1 & 2.

**Evaluation**  We assess the cluster separation of DICE-derived embeddings against a PCA baseline that projects expression profiles to the same latent dimension used for clustering. Qualitatively, we visualize UMAPs (McInnes et al., 2018) (Fig. 1); quantitatively, we report the average *cosine* silhouette score (Rousseeuw, 1987), and cLISI (Korsunsky et al., 2019) computed using the true cluster labels of the test data (Table 1). Because ground-truth labels are available in this benchmark, both metrics are directly comparable. Across all four settings, DICE yields clearer separation in UMAP and better scores than the PCA baseline, indicating more faithful recovery of the underlying classes.

**Uncertainty in the embeddings**  We visualize our confidence set construction for two fixed points: (i) the center of Cluster 2 and (ii) the midpoint between the two cluster centers. For each input, we run DICE 500 times and project the resulting embeddings onto the same UMAP as the training data. When the input lies at the center of a cluster (Figure 2, left), all embeddings map consistently to Cluster 2, indicating high confidence in the cluster assignment. In contrast, when the input lies between clusters (Figure 2, right), the embeddings are split across both clusters, reflecting uncertainty in the assignment. Such uncertainty could be advantageous for downstream tasks designed to incorporate soft labels. Additional plots in Appendix E show how the parameter $\rho$ directly controls the size of the confidence sets. Thus, $\rho$ must be tuned to balance coverage and performance.

## 5 Evaluation on single-cell data

We evaluate the effectiveness of DICE in denoising gene expression profiles using two publicly available single-cell RNA-seq datasets: the CITE-seq dataset from Hao et al. (2021) and the *human fetal brain development* datasets from Polioudakis et al. (2019) and Nowakowski et al. (2017) (see Appendix G for dataset provenance). These datasets originate from distinct tissues and capture diverse cellular populations. They also differ in their relative signal strengths, allowing us to examine the ability of DICE to handle both complex cell distributions and varying signal-to-noise regimes.

**Diffusion Training**  For each dataset, we first select the latent dimension $k$ using the elbow of the singular-value spectrum, and then project the training data onto this $k$-dimensional space via PCA. This yields the training embeddings $\{\widehat{U}_i\}_{i=1}^m$ and the loading matrix $\widehat{V}$ used by DICE. We train the same diffusion architecture across datasets using AdamW with a cosine-annealed learning-rate schedule. Additional details on the training pipeline, including precise model architecture, are provided in Appendix C. We evaluate the influence of picking the PCA dimension ($k$) in Appendix F.

For CITE-seq, training takes about 36 minutes and inference takes about 12 minutes on a setup accelerated by an NVIDIA RTX PRO 6000 GPU. We provide more details on our setup in Appendix C.

### 5.1 Analysis of the CITE-seq dataset

**Dataset**  The CITE-seq dataset (Hao et al., 2021) consists of RNA profiles for 20,729 genes across 152,094 PBMCs with paired *antibody-derived tags* (ADT) measurements. We focus solely on the transcriptomic modality and uniformly subsample 10,000 cells for analysis. Ground-truth labels are provided, and we adopt the *L2*-level granularity, which distinguishes ∼30 immune subtypes.

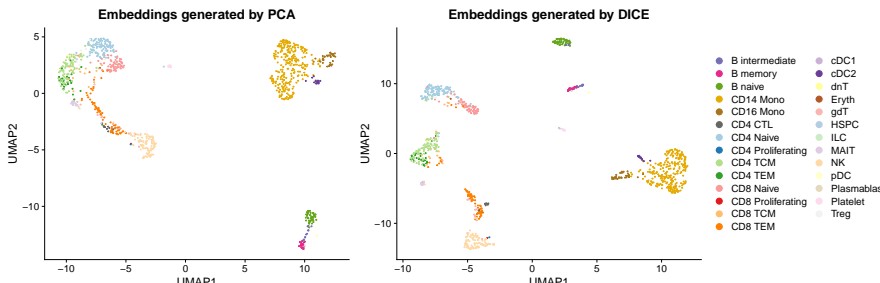

Figure 3: UMAP of 1,000 held-out PBMCs from the CITE-seq dataset. Left: PCA embeddings in a 25-dimensional latent space using $\widehat{V}$ from the training set. Right: embeddings after denoising with DICE using a diffusion model trained on the other 9,000 cells.

| Method | CITE-seq | | | | Human Neo-Cortex | | | |
|---|---|---|---|---|---|---|---|---|
| | ARI | cLISI | NMI | V Measure | ARI | cLISI | NMI | V Measure |
| DICE | **0.805** | 1.344 | **0.740** | **0.813** | **0.393** | **1.406** | **0.553** | **0.559** |
| PCA | 0.745 | **1.334** | 0.689 | 0.787 | 0.347 | 1.496 | 0.496 | 0.525 |
| ALRA | 0.604 | 1.361 | 0.713 | 0.775 | 0.310 | 1.526 | 0.474 | 0.512 |
| kNN (5 nearest neighbors) | 0.713 | **1.334** | 0.647 | 0.756 | 0.275 | 1.837 | 0.349 | 0.405 |
| kNN (10 nearest neighbors) | 0.705 | 1.385 | 0.663 | 0.757 | 0.285 | 1.750 | 0.440 | 0.451 |
| kNN (15 nearest neighbors) | 0.735 | 1.371 | 0.683 | 0.766 | 0.268 | 1.701 | 0.442 | 0.450 |
| MAGIC | 0.674 | 1.397 | 0.648 | 0.757 | 0.317 | 1.451 | 0.502 | 0.509 |
| NMF | 0.448 | 2.332 | 0.430 | 0.556 | 0.209 | 2.237 | 0.220 | 0.291 |
| scVI (10 latent factors) | 0.641 | 1.361 | 0.595 | 0.715 | – | – | – | – |

Table 2: Comparison of clustering metrics across single-cell datasets.

**Preprocessing and denoising** We applied standard QC and pre-processing to the raw UMI counts data (see details in Appendix H), and considered the top 3,000 highly variable genes (Seurat v3 criterion) to construct the data matrix $X \in \mathbb{R}^{10,000 \times 3,000}$. We split into 9,000 training and 1,000 held-out test cells with preserved label proportions. Similar to **Setup 1**, we test DICE's ability in denoising beyond the quality of the training dataset in this setting. The training set was used to fit the diffusion model, and the test set was reserved for evaluation.

Using the training pipeline above, we picked $k = 25$ and trained a diffusion model on 25d PCA embeddings and applied DICE for $T = 100$ denoising iterations to the PCA embeddings of the test cells. We used a linearly decreasing (equally spaced) annealing schedule $\{\rho_t\}_{t=1}^{T}$ from 5 to 0.5 over 100 points. Further details of the denoising procedure are provided in Appendix C.

**Evaluation** We compare UMAP (McInnes et al., 2018) visualizations of the 1,000 held-out test cells denoised by DICE against their PCA projections without denoising (Figure 3). Denoised embeddings show clearer segregation of immune subtypes, with marked improvement in separating sublineages of CD4 and CD8 T cells. The DICE atlas also performs better in segregating the MAIT cells from the other T cells. These populations are notoriously difficult to resolve from RNA alone (Hao et al., 2021), a challenge that originally motivated multimodal approaches combining RNA and protein. In the unimodal setting considered here, however, popular toolkits such as Seurat rely directly on PCA embeddings. Our results indicate that denoising PCA embeddings with priors learned from high-signal training data substantially improves cluster separation, supporting more reliable annotation of held-out cells. We additionally benchmarked the DICE embeddings against several widely used denoising procedures for RNA-seq analysis—MAGIC (van Dijk et al., 2018a), ALRA (Linderman et al., 2022a), $k$-nearest-neighbor smoothing (used in Seurat v3 (Stuart et al., 2019)), and non-negative matrix factorization (NMF)–based embeddings,[4] using the clustering metrics ARI, NMI, cLISI, and V-measure described in Section D. The results (Table 2) show that DICE consistently outperforms these popular denoising pipelines across most clustering metrics.

---

[4]See Appendix I for details on the benchmarking procedures and their implementation.

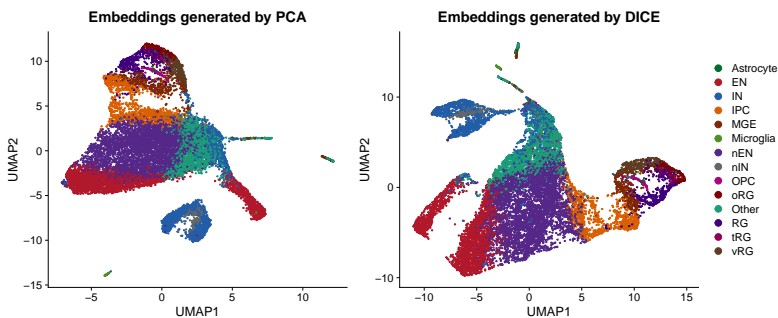

Figure 4: UMAP of 15,126 cells from the Polioudakis et al. (2019) dataset. Left: PCA embeddings in a 15-dimensional latent space using $\widehat{V}$ from the training set. Right: embeddings after denoising with DICE using a diffusion model trained on the Nowakowski et al. (2017) dataset.

## 5.2 ANALYSIS OF HUMAN FETAL BRAIN DEVELOPMENT DATASETS

**Datasets**  We evaluate the effectiveness of DICE in transferring cell type information learned from high-signal training data to related low-signal test data using scRNA-seq datasets from Nowakowski et al. (2017) and Polioudakis et al. (2019). Both datasets profile human fetal brain tissue during development. The Nowakowski et al. (2017) dataset includes cells from primary cortical and *medial ganglionic eminence* (MGE) samples across multiple development stages, whereas the Polioudakis et al. (2019) dataset focuses on cells from the neocortex during mid-gestation. While the cell types profiled in the two datasets are related, they are not identical due to differences in the sampled tissue. This contrasts with the CITE-seq dataset considered previously, where training and test cells originated from the same source. This experiment evaluates the robustness of DICE to realistic distributional changes that occur during cross-dataset cell-type label transfer.

**Preprocessing and Denoising**  We analyzed 3,495 cells from Nowakowski et al. (2017) and 15,126 cells from Polioudakis et al. (2019). After performing standard pre-processing on the raw expression data (see Section H for details), we selected 785 highly variable genes shared by both datasets. This yielded matrices $X_{\mathrm{now}} \in \mathbb{R}^{3495 \times 785}$ and $X_{\mathrm{pol}} \in \mathbb{R}^{15126 \times 785}$. As $X_{\mathrm{now}}$ exhibited a stronger signal, we trained the diffusion prior on a $k = 15$ latent projection of it (via PCA) and evaluated DICE on 1,000 randomly sampled cells from $X_{\mathrm{pol}}$ to assess denoising and embedding quality. The training and denoising hyperparameters (noise/annealing schedule, number of diffusion denoising steps, and number of Gibbs iterations) matched those used in the CITE-seq example.

**Evaluation**  We compare UMAP visualizations for the embeddings constructed from the test data using DICE and the embeddings obtained via the PCA transform learned from the training data. Denoising yielded substantially improved biological interpretability, producing more coherent and homogeneous clusters of similar cell types. A notable improvement is found in the embeddings of the cells belonging to the canonical excitatory trajectory $\mathrm{RG} \rightarrow \mathrm{IPC} \rightarrow \mathrm{nEN} \rightarrow \mathrm{EN}$. This trajectory is visually continuous and easy to follow in the DICE embedding, whereas it appears fragmented and noisy under PCA. Similar to the CITE-seq example, we benchmarked the DICE embeddings against MAGIC, ALRA, $k$-nearest smoothing, and NMF, using the same set of clustering metrics. The results summarized in 2, show that DICE consistently outperforms all competing denoising methods across all four metrics, validating our claim that diffusion-based denoising sharpens lineage relationships and resolves biologically relevant subpopulations better than existing techniques.

**Discussion**  We introduced DICE, a latent plug-and-play framework for denoising and extracting meaningful embeddings from high-dimensional observational data with an underlying low-rank structure. Both synthetic and real-world experiments demonstrated its ability to denoise beyond the training distribution and the ability to leverage clean reference data for denoising. We further showed that DICE can quantify uncertainty in cluster assignments and remains robust under model misspecification. Future work includes 1) extending DICE beyond linear low-rank structures and the i.i.d. noise assumption, 2) improving the efficiency of the sampling procedure, 3) incorporating multimodal data and spatial context, and 4) evaluating the quality of the produced embeddings on clinically meaningful downstream tasks.

**Reproducibility statement**   We provide the full codebase as supplementary material and will release it publicly upon publication. The appendix details the implementation, including hardware specifications, software packages, and training configurations. For synthetic experiments, we include both a description in the main text and code for data generation and preprocessing in the supplementary files. For real-world datasets, we describe dataset acquisition in the appendix, preprocessing steps in both the main text and appendix, and provide scripts to reproduce all experiments.

**Use of Large Language Models (LLMs)**   Large Language Models (LLMs) were used to assist in preparing tables and figures and for proofreading.

**Acknowledgments**   This work was supported in part by an AWS Cloud Credit Grant from Cornell's Center for Data Science for Enterprise and Society.

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

## A    DIFFUSION TRAINING

For a given noise schedule $\{\alpha_t\}_{t \in [T]}$, noisy versions of $\widehat{U}_i^{(r)}$ are generated as

$$\widehat{U}_{t,i}^{(r)} = \sqrt{\bar{\alpha}_t}\,\widehat{U}_i + \sqrt{1-\bar{\alpha}_t}\,\varepsilon_{t,i}, \qquad \varepsilon_{t,i} \overset{\text{i.i.d.}}{\sim} \mathcal{N}_k(0, I_k), \tag{8}$$

where $\bar{\alpha}_t = \prod_{s=1}^t \alpha_s$. A neural network $\widehat{\varepsilon}_{\theta_t}(\widehat{U}_{t,i}^{(r)}; t)$ is trained to predict the injected noise $\varepsilon_{t,i}$ from the corrupted sample $\widehat{U}_{t,i}^{(r)}$ by minimizing the mean-squared error

$$\mathcal{L}(\theta) = \frac{1}{m} \sum_{i=1}^m \left\| \varepsilon_{t,i} - \widehat{\varepsilon}_\theta(\widehat{U}_{t,i}^{(r)}; t) \right\|_2^2.$$

The fitted network provides an estimate of $\mathbb{E}[\varepsilon_{t,i} \mid \widehat{U}_{t,i}^{(r)}]$, which is directly related to the score function of the marginal distribution $p_t(\widehat{U}_{t,i}^{(r)})$ via

$$\nabla_{\widehat{U}_{t,i}^{(r)}} \log p_t(\widehat{U}_{t,i}^{(r)}) \approx -\frac{1}{\sqrt{1-\bar{\alpha}_t}}\,\widehat{\varepsilon}_\theta(\widehat{U}_{t,i}^{(r)}; t).$$

Thus, training the model to predict noise is equivalent to estimating the score function, which in turn defines the reverse diffusion process and enables sampling from the prior $P_{\text{prior}}$. For the detailed procedure, see Algorithm 2.

## B    PROOF OF PROPOSITION 3.1

To prove Proposition 3.1 let us observe that if $f(\cdot)$ is the density of standard multivariate Gaussian distribution, then

$$f(X_i - VZ_i) = \frac{1}{(2\pi)^{d/2}} \exp\left( -\frac{1}{2} \|X_i - VZ_i\|_2^2 \right).$$

Therefore

$$P_\rho(Z_q^{(s)} \mid X_q, U_q) \propto \exp\left( -\frac{1}{2}\|X_q - \widehat{V}Z_q^{(s)}\|_2^2 - \frac{1}{2\rho_s^2}\|U_q^{(s)} - Z_q^{(s)}\|_2^2 \right) \tag{9}$$

$$\overset{(1)}{\propto} \exp\left( -\frac{1}{2}(Z_q^{(s)})^\top \left( \frac{1}{\rho_s^2}I_k + \widehat{V}^\top \widehat{V} \right) Z_q^{(s)} + (Z_q^{(s)})^\top \left( \widehat{V}^\top X_q + \frac{1}{\rho_s^2}U_q^{(s)} \right) \right) \tag{10}$$

$$\overset{(2)}{\propto} \exp\left( -\frac{1}{2}(Z_q - m_q)\Lambda^{-1}(Z_q - m_q) \right), \tag{11}$$

where

$$\Lambda = \left( \frac{1}{\rho_s^2}I_k + \widehat{V}^\top \widehat{V} \right)^{-1} \tag{12}$$

$$m_q = \Lambda \left( \widehat{V}^\top X_q + \frac{1}{\rho_s^2}U_q^{(s)} \right), \tag{13}$$

and (2) follows by completing the quadratic form in the power of the exponential in (1) to match the density of a Gaussian distribution. Then the proposition follows by identifying the density on the right-hand side of (2) as that of a Gaussian distibution with mean $m_q$ and covariance $\Gamma$.

---

**Algorithm 2** Train diffusion model

---

**Require:** Training set $\{X_i^{(r)}\}_{i \in [T]}$, epochs $E$, noise schedule $\{\alpha_t\}_{t \in [T]}$, embedding dimension $k$
**Ensure:** Trained model $\widehat{\varepsilon}_\theta$

1: Calculate the factor loading matrix $\widehat{V}$ from the training data by projecting $X^{(r)}$ to a $k$ dimensional latent space using PCA.
2: Compute $\widehat{U}_i = \widehat{V}^\top X_i^{(r)}$.
3: **for** $e = 1, \ldots, E$ **do**
4:     Independently sample timestep $t^{(i)} \sim \mathcal{U}\{1, \ldots, T\}$ for all $i = 1, \ldots, m$
5:     Draw noises $\varepsilon_{t^{(i)}, i} \sim \mathcal{N}(0, 1)$ for all $i$
6:     Construct noised inputs

$$\widehat{U}_{t,i}^{(r)} = \sqrt{\bar{\alpha}_{t^{(i)}}} \, \widehat{U}_i + \sqrt{1 - \bar{\alpha}_{t^{(i)}}} \, \varepsilon_{t^{(i)}, i}, \qquad \text{for } i = 1, \ldots, n,$$

and $\bar{\alpha}_{t^{(i)}} = \prod_{s=1}^{t^{(i)}} \alpha_s$
7:     Compute loss

$$\mathcal{L}(\theta) = \frac{1}{m} \sum_{i=1}^{m} \left\| \varepsilon_{t,i} - \varepsilon_\theta\big(\widehat{U}_{t,i}^{(r)}, t^{(i)}\big) \right\|_2^2$$

8:     Take gradient step on $\theta$.
9: **end for**
10: **return** $\varepsilon_\theta$

---

## C   MODEL ARCHITECTURE AND IMPLEMENTATION DETAILS FOR TRAINING THE DIFFUSION MODELS USED IN THE SINGLE CELL EXPERIMENTS

**Overview** Across all experiments, we use the same denoising network, `TabularDiffusionMLP`, which predicts Gaussian noise $\varepsilon_{t,i}$ from the noised sample $\widehat{U}_{t,i}^r \in \mathbb{R}^k$ at diffusion step $t$. The architecture is parameterized by the number of residual MLP blocks $M$ and the hidden dimension $D$. For synthetic experiments, we use a smaller model with $M = 2$ blocks and hidden dimension $D = 64$ ($\approx 150{,}000$ trainable parameters), trained for $E = 2{,}000$ epochs. For single-cell experiments, we employ a larger model with $M = 8$ blocks and hidden dimension $D = 512$ ($\approx 35{,}000{,}000$ trainable parameters), trained for $E = 10{,}000$ epochs. We use a batch size of $B = 4048$ across all models.

**Input encoders** We encode the data and the diffusion time with separate branches: (i) a linear projection $U_t^r \mapsto \mathbb{R}^D$, and (ii) a sinusoidal positional embedding of $t$ followed by two SiLU-activated linear layers to $\mathbb{R}^D$. We then concatenate the two $D$-dimensional features into a $2D$-dimensional vector.

**Backbone** The concatenated features are processed by eight residual MLP blocks of constant width $2D$. Each block expands to $4D$ units and contracts back to $2D$ (Linear $2D \to 4D$, Batch-Norm1d, SiLU, Linear $4D \to 2D$, BatchNorm1d), and adds a residual skip connection. This design provides sufficient capacity while remaining simple and fast for tabular inputs.

**Output head** A final projection (Linear $2D \to D$, SiLU, Linear $D \to d$) produces $\varepsilon_\theta(\widehat{U}_t^{(r)}, t)$ in the same dimensionality as $\widehat{U}_t^{(r)}$.

**Training objective and schedule** We adopt the standard noise-prediction loss $\mathcal{L}(\theta)$ defined in Algorithm 2 with a linear $\beta$ schedule: $\beta_t \in [10^{-4}, 2 \times 10^{-2}]$ linearly spaced over $T = 512$ steps and $\alpha_t = 1 - \beta_t$, $\bar{\alpha}_t = \prod_{s=1}^{t} \alpha_s$. We optimize with AdamW (learning rate $1 \times 10^{-4}$, batch size 4048) for 20000 epochs.

**Data augmentation in training** High-quality single-cell datasets may contain relatively few cells but provide reliable labels. We leverage these labels for data augmentation during diffusion model training. Inspired by *mixup* by Zhang et al. (2018), we interpolate between multiple same-class

samples rather than mixing across classes. Concretely, given a sample $X_i^{(r)}$ with label $Y_i^{(r)}$, we select four additional same-class points and construct

$$\tilde{X}_i^{(r)} = \sum_{j=1}^{5} \lambda_j x_j, \quad (\lambda_1, \ldots, \lambda_5) \sim \text{Dirichlet}(1).$$

During training, we select the interpolated version $\tilde{X}_i^{(r)}$ with a probability of $p = 0.9$. This intra-class mixup enriches the training distribution and improves robustness for the single-cell data. We use no data augmentation for the synthetic experiments.

**Layer specification**  For completeness, Table 3 lists the exact layers and tensor shapes ($B$ denotes batch size).

| Stage | Operation / Activation | Output shape |
|---|---|---|
| *Input projections* | | |
| $\widehat{U}_t^{(r)}$ branch | Linear $(k \rightarrow D)$ | $(B, D)$ |
| $t$ embedding | Sinusoidal $(1 \rightarrow 16)$ | $(B, 16)$ |
| | Linear $(16 \rightarrow D)$ + SiLU | $(B, D)$ |
| | Linear $(D \rightarrow D)$ + SiLU | $(B, D)$ |
| CONCAT | — | $(B, 4D)$ |
| *Residual MLP block* (repeated $\times M$) | | |
| Hidden | Linear $(2D \rightarrow 4D)$ | $(B, 4D)$ |
| | BatchNorm1d $(4D)$, SiLU | $(B, 4D)$ |
| | Linear $(4D \rightarrow 2D)$ | $(B, 2D)$ |
| | BatchNorm1d $(2D)$ | $(B, 2D)$ |
| Residual | $x \leftarrow x + \text{block}(x)$ | $(B, 2D)$ |
| *Output projection* | | |
| Head | Linear $(2D \rightarrow D)$ + SiLU | $(B, D)$ |
| | Linear $(D \rightarrow k)$ | $(B, k)$ |

Table 3: Layer specification for `TabularDiffusionMLP` predicting $\varepsilon_\theta(\widehat{U}_t^{(r)}, t)$. $B$ is batch size, $k$ is the input (latent) dimension, $D$ is the hidden dimension of the network

**Practical notes**  We normalize $t$ to $[0, 1]$ prior to sinusoidal embedding, use SiLU activations throughout, BatchNorm1d within blocks, and default PyTorch initializations.

**Implementation setup**  We implemented all experiments in Python. Neural network training was performed in PyTorch (Paszke et al., 2019) on GPU-accelerated hardware, using a mix of AWS `g4dn.xlarge` EC2 instances and a dedicated Ubuntu server with an NVIDIA RTX PRO 6000 GPU. Our whole code is available online at

**Runtime**  The runtime of our method can be divided into training and inference.

*Training* is dominated by parameter updates for the diffusion model, with speed depending on hardware and chosen model size. We report the execution times for the Ubuntu server with an NVIDIA RTX PRO 6000 GPU. Using the CITE-seq setup, training on 9,000 cells with 3,000 dimensions for 10,000 epochs required approximately 36 minutes.

*Inference* proceeds via Gibbs sampling, alternating between the prior and likelihood steps. The prior step, which involves sampling from the diffusion model, is the most computationally expensive. As $\rho$ determines the starting timestep of the diffusion model $t$, the duration of the prior sampling is dependent on $\rho$, larger values of $\rho$ result in longer runtime. The total runtime for denoising the test set of the CITE-seq dataset with 100 Gibbs iterations and $\rho$ ranging from 5.0 to 0.5 took 12 minutes.

## D  DISCUSSION ON THE CLUSTERING METRICS

To assess the quality of clustering in the learned embeddings, we employ five complementary metrics: the average cosine silhouette score (Rousseeuw, 1987), the Adjusted Rand Index (ARI) (Hubert & Arabie, 1985), the cell-type Locally Invariant Simpson Index (cLISI) (Korsunsky et al., 2019), the Normalized Mutual Information (NMI) (Strehl & Ghosh, 2002), and the V-measure (Rosenberg & Hirschberg, 2007). Below, we provide a brief description of each metric.

1. **Average Cosine Silhouette Score.** The silhouette score quantifies how well a point is matched to its assigned cluster compared to other clusters, measured here using cosine distance. For a point $i$, let $a(i)$ denote the average intra-cluster cosine distance, and let $b(i)$ denote the minimum average cosine distance between $i$ and any other cluster. The silhouette score for $i$ is defined as

$$s(i) = \frac{b(i) - a(i)}{\max\{a(i), b(i)\}}.$$

   We report the mean silhouette score across all points as a global measure of clustering quality, with values closer to 1 indicating more distinct and coherent clusters.

2. **Adjusted Rand Index (ARI).** The Rand Index evaluates the agreement between two partitions by counting the proportion of point pairs that are consistently assigned together or apart. The ARI corrects this measure for chance, making it more robust in settings with many clusters. It is defined as

$$\text{ARI} = \frac{\text{RI} - \mathbb{E}[\text{RI}]}{\max(\text{RI}) - \mathbb{E}[\text{RI}]},$$

   where RI is the raw Rand Index and $\mathbb{E}[\text{RI}]$ is the expected value of the number of agreeing pairs under random clustering but fixed cluster sizes. ARI values range from 0 (chance-level agreement) to 1 (perfect alignment with ground truth).

3. **Cell-type Locally Invariant Simpson Index (cLISI).** The Local Inverse Simpson's Index (LISI) was originally introduced to assess batch mixing in single-cell integration tasks. We adapt it to clustering by replacing batch labels with cluster (or cell-type) labels, yielding cLISI. For each cell $i$, cLISI measures the effective number of distinct clusters represented in its $k$-nearest neighbor neighborhood:

$$\text{LISI}(i) = \left( \sum_c p_{ic}^2 \right)^{-1},$$

   where $p_{ic}$ is the fraction of neighbors of $i$ belonging to cluster $c$. We report the average cLISI across all cells. Lower values correspond to locally purer clusters, while higher values indicate greater mixing. Unlike ARI, cLISI does not require ground truth annotations and provides a local measure that complements the global silhouette score.

4. **Normalized Mutual Information (NMI).** Normalized Mutual Information is an information-theoretic measure for computing the agreement between two sets of cluster labels for the same data points. It measures the agreement between two partitions of the data, $C$ and $T$ as follows:

$$\text{NMI}(C, T) = \frac{2\, I(C; T)}{H(C) + H(T)},$$

   where $I(C; T)$ is the mutual information between $C$ and $T$, and $H(\cdot)$ denotes the Shannon entropy (see, Cover & Thomas (2006)).

   The NMI ranges from 0 to 1, with larger values indicating stronger agreement between the inferred clusters and the biological ground-truth labels. It provides a global summary of clustering accuracy and is widely used for benchmarking atlas-level annotation quality.

5. **V-measure.** The V-measure provides an entropy-based evaluation scheme of two sets of cluster labels. In our case, these sets are the cluster labels obtained by clustering the denoised embeddings and the original cell-type metadata. The V-measure computes two

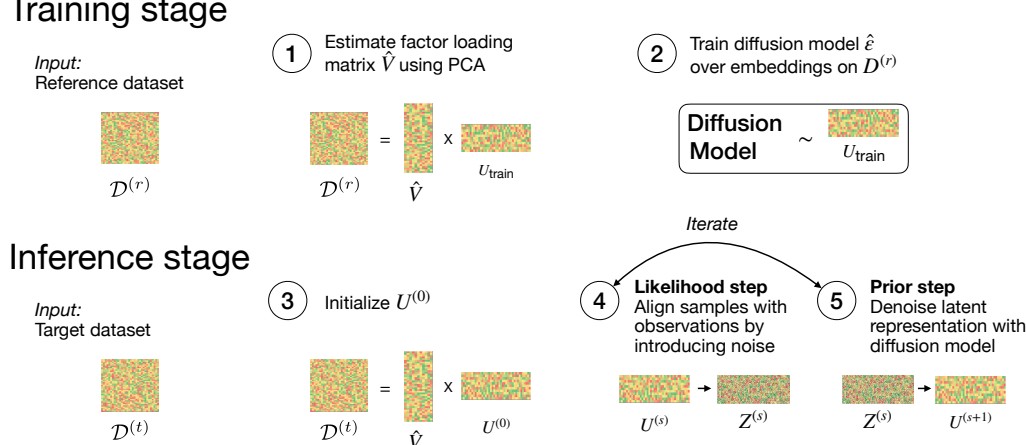

Figure A.1: Visual sketch of the DICE denoising procedure. Training stage: (1) The factor loading matrix $\hat{V}$ is estimated on the reference dataset. (2) A diffusion model is trained to capture the underlying biological variation. Inference stage: (3) The initial latent representation $U^{(0)}$ is obtained using $\hat{V}$. Iteratively: (4) Noise is added to $U^{(s)}$ to produce $Z^{(s)}$. (5) $Z^{(s)}$ is denoised using the learned diffusion model, yielding $U^{(s+1)}$.

quantities, namely, homogeneity $h$ and completeness $c$. Let $K$ be the true cell labels and $C$ be the predicted clusters. Then

$$h = 1 - \frac{H(K \mid C)}{H(K)}, \quad \text{and} \quad c = 1 - \frac{H(C \mid K)}{H(C)},$$

where $H(\cdot)$ refers to the Shannon entropy of a distribution. Then, the V-measure is defined as

$$\text{V-measure} = \frac{2 \times h \times c}{h + c}.$$

Higher values of V-measure indicate that the reconstructed atlas adheres to the cluster structure encoded in the metadata.

## E    SENSITIVITY TO $\rho$

DICE employs a split Gibbs sampling scheme that introduces an auxiliary variable $Z_i$ to decouple the likelihood and prior updates. The strength of the coupling between $Z_i$ and $U_i$ is controlled through the Gaussian alignment penalty

$$-\frac{1}{2\rho^2}\|U_i - Z_i\|_2^2, \tag{14}$$

which appears in both the likelihood and prior steps (see Equation 5 and Equation 6). Intuitively, $\rho$ controls the distance between the sampling of $U_i$ and $Z_i$ in both the likelihood and prior step.

To interpret the role of $\rho$, we start by reviewing the canonical inverse problem of denoising: we observe a noisy matrix $X$, and our goal is to recover a clean, underlying matrix $UV^\top$, where $U$ is unknown. This problem is naturally framed in a Bayesian setting. The likelihood $f(X - UV^\top \mid U)$, models the noise process, while the prior $P_{\text{prior}}(U)$ encodes our assumptions about the structure of $U$. A standard approach is to find the *Maximum a Posterior* (MAP) estimate, which solves the optimization problem:

$$\widehat{U}_{\text{MAP}} = \arg\min_U \underbrace{-\log f(X - UV^\top \mid U)}_{\ell(U;X)} \underbrace{-\log P_{\text{prior}}(U)}_{R(U)},$$

where $\ell(U; X)$ is the data-fidelity term, and $R(U)$ is the regularizer induced by the prior. While this yields a single point estimate, it discards all uncertainty about the solution.

To capture the full posterior distribution, we turn to *posterior sampling*, targeting:

$$\pi(U \mid X) \; \propto \; f\big(X - UV^\top \,\big|\, U\big) \; P_{\text{prior}}(U).$$

This problem can be viewed as a *soft relaxation* of the MAP problem. The key is to observe that sampling from this posterior can be efficiently achieved using a Split Gibbs Sampler. (Xu & Chi, 2024; Venkatakrishnan et al., 2013b).

The sampler introduces an auxiliary variable $Z$ and iterates between two conditional sampling steps:

1. **Data-consistency Step (likelihood step):** Sample $Z$ from $p(Z \mid U, X)$, which typically involves the likelihood $f$ and projects information from the observation $X$ and the current state $U$.

2. **Prior-projection Step (prior step):** Sample $U$ from $p(U \mid Z) \propto P_{\text{prior}}(U) \cdot \delta(U \approx Z)$. This critical step draws a sample from the prior distribution that is consistent with the auxiliary variable $Z$.

As we repeat this sampling procedure, the two steps eventually reach a stochastic equilibrium. When $\rho$ is small, the strong coupling term dominates both steps. In this regime:

- The initialization (typically from the data distribution as in Algorithm 1) has a persistent influence because the strong coupling prevents substantial deviation from previous states.
- Both the likelihood and prior terms become relatively less influential compared to the alignment constraint.
- The sampler produces outputs that remain close to the initialization throughout the sampling process.

When $\rho$ is large, the coupling is weak, allowing:

- The prior term to dominate the U-update step, pulling samples toward the prior distribution.
- The data likelihood to have more influence in the Z-update step.
- Greater exploration away from the initialization.

Overall, $\rho$ controls the trade-off between fidelity to the observation and adherence to the learned prior. Small $\rho$ enforces tight coupling and rapid convergence near the data distribution, while large $\rho$ allows freer interaction between the two updates and greater influence of the prior. In the following experiments, we empirically investigate how varying $\rho$ affects the resulting embeddings when (i) applying DICE repeatedly to the same input observation (subsection E), and (ii) when the clusters present in the test data differ from those seen during training (subsection E.2).

### E.1 SPREAD ACROSS RUNS

**Setup** To assess the sensitivity of DICE to different values of $\rho$, we perform an ablation study by denoising a fixed input point multiple times and comparing the spread across runs. By fixing the input observation to a single point, we can isolate the effect of $\rho$ on the variability introduced by the stochastic sampling procedure. We use the two-cluster setup (see section 4, Setup 1), and use the same input point for all runs, the center of Cluster 2. We generate 500 independent runs of DICE for $\rho \in \{0.1, 0.5, 1.0, 5.0, 10.0, 20.0\}$.

**Results** We visualize the generated embeddings under different values of $\rho$ with a UMAP in Figure A.2. As $\rho$ increases, the spread of the sampled points grows: for small $\rho$, points remain tightly concentrated around the cluster center, while for large $\rho$, they disperse more broadly. The spread of the sampled points can be interpreted as the posterior uncertainty under different assumed noise levels. When the assumed noise is low (small $\rho$), the posterior is sharply concentrated around the observation, as the likelihood dominates and little deviation from the measurement is plausible. In contrast, for higher noise levels (large $\rho$), the posterior becomes broader, reflecting greater uncertainty: a wider range of latent points is consistent with the observed data. Thus, varying $\rho$ calibrates how strongly the likelihood influences the denoised embedding.

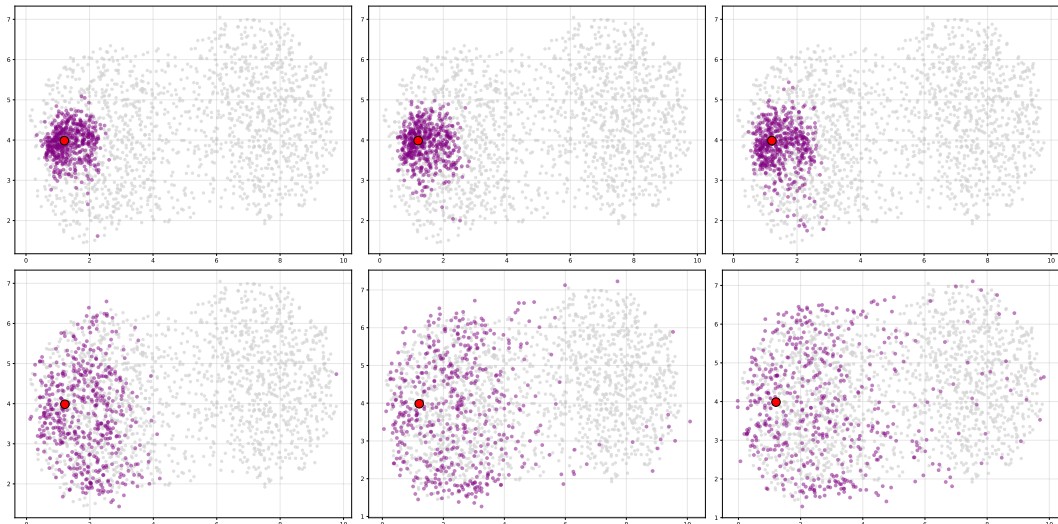

Figure A.2: UMAP visualizations of 500 runs of DICE-denoised embeddings for the Cluster 2 center under Setup 1 with increasing $\rho \in [0.1, 0.5, 1.0, 5.0, 10.0, 20.0]$ (left to right, top to bottom). Input observation in red; training data in grey.

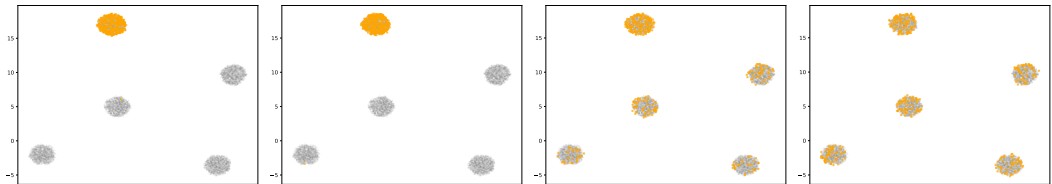

Figure A.3: UMAP visualizations of PCA projections (left) and DICE-denoised embeddings for test points restricted to a single cluster, with increasing values of $\rho \in [1.0, 8.0, 20.0]$ (left to right). As $\rho$ increases, the embeddings gradually shift from being measurement-aligned to prior-aligned. Training data are shown in grey.

### E.2 MISMATCH BETWEEN TRAINING AND TEST CLUSTERS

In this section, we investigate the influence of prior guidance under different $\rho$. We create a mismatch between the training and the test data, and evaluate whether points stay close to the measurement, or are moved according to the prior. We update our training data fixed to contain 5 well separated clusters. Then, we vary the input test data, which either comes only from one cluster, or from a new cluster which is not present in the training data. Our diffusion model is the same as for the other synthetic experiments (see Appendix C). We run DICE for 200 Gibbs iterations without averaging, and report results for different choices of $\rho$.

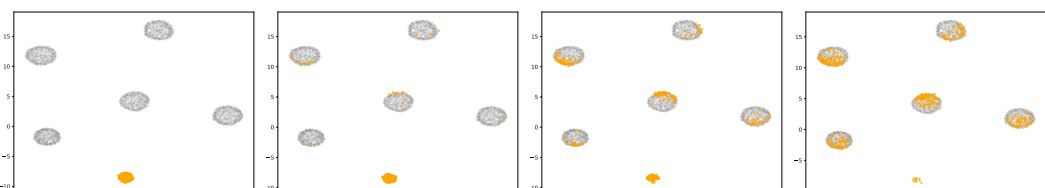

Figure A.4: UMAP visualizations PCA projections (left) and DICE-denoised embeddings for test points from a cluster *not observed during training*, shown for increasing $\rho \in [0.5, 2.0, 4.0]$ (left to right). With larger $\rho$, the unseen cluster progressively gets mapped to the training clusters, reflecting stronger prior alignment. Training data are shown in grey. The UMAP projection is fitted jointly on the combined training and test data.

**Training data**  We extend our synthetic data generation to include five instead of two cluster centers, while still following the same low-rank latent structure. We chose low noise levels, such that the training data is well separated. In detail, we start by drawing the cluster centers $\mu_i \sim \mathcal{N}_{15}(0, I_{15})$ in latent dimension $k = 15$. The training prior is a balanced Gaussian mixture, $P_{\mathrm{prior}} = \sum_i^5 \frac{1}{5} \mathcal{N}_{15}(\mu_i, 0.3\,I_{15})$ with each component corresponding to one cell type. We sample a loading matrix $V \in \mathbb{R}^{200 \times 15}$ with entries i.i.d. $\mathcal{N}(0, 1)$. For each cell $i$, we draw a latent $U_i^{(r)} \sim P_{\mathrm{prior}}$ and measurement noise $\varepsilon_i^{(r)} \sim \mathcal{N}_{200}(0, 0.5I_{200})$, and generate a synthetic expression profile $X_i^{(r)} = VU_i^{(r)} + \varepsilon_i^{(r)} \in \mathbb{R}^{200}$, $i = 1, \ldots, 8000$. This yields a training dataset $\mathcal{D}^{(r)}$ of size $m = 8{,}000$ in $d = 200$ observed dimensions. We generate a test dataset $\mathcal{D}^{(t)}$ of size $m = 2{,}000$ by increasing the noise variance $\varepsilon_i^{(t)} \sim \mathcal{N}_{200}(0, I_{200})$.

### E.2.1 FEWER CLUSTERS IN THE TRAINING DATA

**Setup**  We construct a test set consisting solely of cells from a single cluster. This setup corresponds to a mismatch where the model's learned prior encompasses a richer structure than what is present in the observed data: While DICE is trained on a uniform distribution across clusters, where each cluster is equally likely, the test data only contains one cluster.

**Results**  We plot the PCA projection (left), and generated denoised embeddings for $\rho \in \{1.0, 8.0, 20.0\}$ (left to right) in Figure A.3. DICE projects all samples into one of the learned clusters, independent of $\rho$. However, $\rho$ controls which of the learned clusters the observations are assigned to. For small $\rho$, samples remain close to their noisy measurements, resulting in all embeddings staying in the original input cluster. For intermediate $\rho$, most points concentrate within one cluster, but a subset diffuses into neighboring clusters. For large $\rho$, the denoising process aligns closely with the prior, resulting in approximately uniform assignment across clusters, independent of the original observation.

### E.2.2 MORE CLUSTERS IN THE TRAINING DATA

We now consider the opposite case, where the test data includes a new cluster that was unseen during training. Specifically, we sample an additional cluster center $\mu_i$ from the same generative process as in the main setup, ensuring that this cluster lies outside the support of the training data.

**Results**  We show a UMAP visualization of PCA projection (left), and generated denoised embeddings for $\rho \in \{0.5, 2.0, 4.0\}$ in Figure A.4. To observe the test cluster, the UMAP is fitted jointly on the combined training and test data. The PCA projection (left) shows that test measurements form a new cluster, which is non-overlapping with the initial training clusters plotted in gray. For small $\rho$, DICE maintains a distinct cluster of points around the novel cluster. As $\rho$ increases, samples gradually disperse into the nearest training clusters, reflecting the influence of the prior. With $\rho = 4.0$, almost all points are assigned to clusters from the training set. Notably, even for large $\rho$, the measurements still exert a directional influence on the embedding. Points are not assigned uniformly inside the clusters, but tend to align with parts of the clusters more consistent with the observed input.

### E.3 DISCUSSION

The results obtained by varying $\rho$ highlight the trade-off between data fidelity and prior alignment. A small $\rho$ keeps denoised samples close to their observations, preserving local variations even when they deviate from the training data. In contrast, a large $\rho$ guides samples toward representations consistent with the learned prior. This ability to steer between these regimes makes $\rho$ a powerful control parameter for balancing embedding quality across datasets of differing noise levels. When the data are assumed to be low-noise, a small $\rho$ maintains proximity to the measurements, allowing structure not captured in the training set to emerge. Conversely, for high-noise data, a larger $\rho$ leverages the prior to guide the estimates toward more reliable, denoised representations.

## F  SENSITIVITY TO $k$

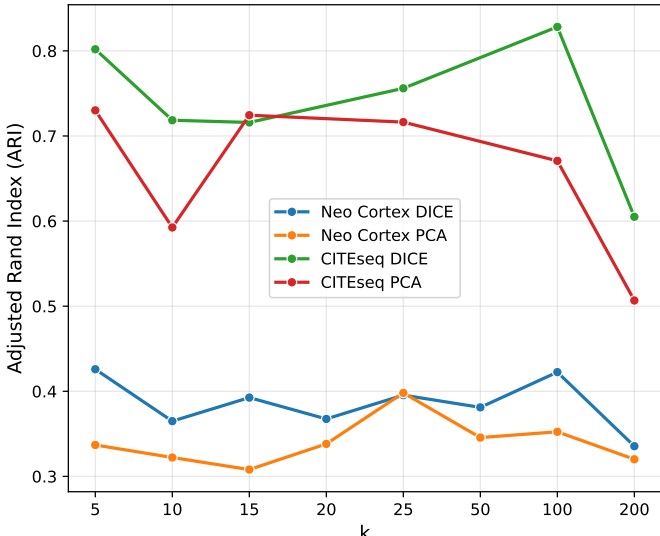

Figure A.5: Analysis of Adjusted Rand Index(ARI) for different values of $k$, the dimension of the latent space. x-axis is not to scale.

**Setup**  To assess the sensitivity of the DICE embeddings to the choice of latent dimension $k$ in Eq. (2), we conducted an ablation study on both the CITE-seq and human neocortex datasets using $k \in \{5, 10, 15, 20, 25, 50, 100, 200\}$. The quality of the resulting embeddings was evaluated using the Adjusted Rand Index (ARI).

**Setup**  We summarize our findings in Figure A.5. For the CITE-seq dataset, the ARI values of the DICE embeddings increase up to $k = 100$ and decreases thereafter. Furthermore, as the latent dimension grows, the benefit of DICE-based denoising over PCA based embeddings becomes more pronounced. This is expected given the large number of distinct cell types in the dataset, which naturally requires a higher-dimensional latent representation. In contrast, for the human neocortex dataset, the ARI values remain relatively stable across different choices of $k$, with DICE outperforming naive PCA-based embeddings for most values. Based on these observations, we recommend selecting $k$ by identifying an elbow in the scree plot of the singular values; choosing a slightly larger $k$ is generally safe, but excessively large values lead to noisier embeddings and increased computational cost with minimal marginal gain.

## G  DATA SOURCES

**CITE-seq benchmark.**  We used the *Seurat v4 CITE-seq* dataset distributed with `scvi-tools` as our atlas building benchmark. The original Seurat object and feature matrices were taken directly from the scvi example and used without modification, except for preprocessing and downsampling steps described in Section 5.

**Human fetal brain development.**  We analyzed two fetal neocortex transcriptomic datasets, Nowakowski et al. (2017) and Polioudakis et al. (2019), obtained via the gEAR (Gene Expression Analysis Resource) portal (Orvis et al., 2021). For both studies, we downloaded the author-provided processed count matrices and accompanying metadata from gEAR and restricted analyses to the shared gene set as detailed in Section 5.

**Provenance and licensing.**  All datasets were used under the terms specified by their original authors and hosting platforms. We performed only secondary analysis; no new data were generated for this work.

## H PREPROCESSING DETAILS FOR THE SINGLE CELL DATASETS

### H.1 CITE-SEQ DATASET

We adopt the following pre-processing pipeline for the CITE-seq data obtained from `scvi-data`. A random stratified sampling was used to subsample 10000 cells from the total pool of cells, maintaining the relative proportion of different cell types.

**QC filtering** To remove the cells with poor expression quality and sequencing errors, we adopt the standard QC scheme adopted in Seurat Stuart et al. (2019) and `scanpy`. We exclude cells with fewer than 200 detected genes and filter out genes expressed in fewer than 3 cells. Additionally, to eliminate cells that may have been dying or stressed at the time of sequencing, we remove all cells with more than 15% mitochondrial RNA content.

**Normalization and Batch Effect Removal.** To account for differences in sequencing depth across cells, we perform standard library-size normalization: for each cell, we divide the gene counts by the total UMI count and multiply by $10^4$. The resulting values are then transformed using $x \mapsto \log(1 + x)$ to stabilize variance and reduce the influence of highly expressed genes. We selected the top 3,000 highly variable genes (Seurat v3 criterion), scaled to unit variance with values capped at 10. Batch effects from sequencing lanes were corrected with Harmony (Korsunsky et al., 2019), yielding an expression matrix $X \in \mathbb{R}^{10000 \times 3000}$.

### H.2 HUMAN NEO-CORTEX DATASETS

We analyzed 3,495 cells from Nowakowski et al. (2017) and 15,126 cells from Polioudakis et al. (2019). The pre-processed datasets were obtained from the gEAR (Gene Expression Analysis Resource) portal (Orvis et al., 2021). The downloaded data had already undergone standard QC procedures, as described for the CITE-seq dataset, and the expression values were normalized to a fixed library size and transformed using the $\log 1p$ operation.

Because the two studies differed in experimental protocols and developmental stages, only 17,638 genes were shared across both datasets. From these, we selected the top 5,000 highly variable genes (HVGs) within each dataset and retained the intersection of these HVG sets, yielding 785 shared highly variable genes. This resulted in the matrices $X_{\text{now}} \in \mathbb{R}^{3495 \times 785}$ and $X_{\text{pol}} \in \mathbb{R}^{15126 \times 785}$.

Cell-type annotations were available only for the Nowakowski et al. (2017) dataset, so we transferred these labels to the Polioudakis et al. (2019) cells using the Seurat v3 label transfer procedure (Stuart et al., 2019) based on the shared set of genes.

## I IMPLEMENTATION DETAILS FOR BENCHMARKING

To benchmark DICE embeddings against those generated by popular techniques for scRNA-seq analysis, we compared the performance of our method against *MAGIC* (van Dijk et al., 2018b), *ALRA* (Linderman et al., 2022b), *NMF (Non-negative Matrix Factorization)*, *scVI* (Lopez et al., 2018a) and $k$-nearest neighbor based denoising with $k = 5, 10$ and 15. We evaluated clustering performance using Seurat's FindClusters algorithm with the Leiden method (Traag et al., 2019). To select the optimal hyperparameter, we ran the algorithm over a range of resolution values (0.01–2.0) and retained the clustering that achieved the highest ARI. In the following, we summarize the procedures and their implementation inwour analysis.

- *MAGIC* implements a Markov affinity based diffusion smoothing of cellular expression profiles.In our benchmarking experiment, we computed PCA scores from the expression matrix and supplied them directly to MAGIC for denoising. We used the official implementation available at `https://github.com/KrishnaswamyLab/MAGIC`.
- *ALRA* applies an adaptively thresholded low-rank matrix approximation to recover denoised gene-expression values. We employed the *pyALRA* Python implementation (Lanau & Waterfall, 2025) for all experiments. The expression profiles were provided to ALRA and a low rank approximation was computed. We used 50 PC scores of the denoised matrix as our embeddings for the nechmarking.

- *Non-negative Matrix Factorization (NMF)* is a commonly used alternative to PCA for denoising and latent factor extraction in scRNA-seq analysis (Kotliar et al., 2019; Carbonetto et al., 2026). We used the NMF implementation from *scikit-learn* (Pedregosa et al., 2011), selecting the latent dimension to match the number of PCs used for the DICE embeddings.

- *scVI* is a variational-autoencoder–based model that denoises raw UMI counts while accounting for library-size variation and batch effects. For the CITE-seq dataset, we trained scVI with a 10-dimensional latent spacewand used the estimated posterior expectations of the latent variables as our embeddings. We could not apply scVI to the human fetal neocortex dataset because the raw UMI counts were not available.

- $k$-*nearest-neighbor smoothing* is a classical local-averaging denoising strategy widely used in pipelines such as Seurat v3 (Stuart et al., 2019). We denoised PCA embeddings by replacing each cell's representation with the average of its $k$ nearest neighbors (Euclidean distance), using $k \in \{5, 10, 15\}$.

