# OpenReview forum: "Clustering by Denoising: Latent plug-and-play diffusion for single-cell embeddings"
_ICLR.cc/2026/Conference — ICLR 2026 Poster_

### Official Review · Reviewer_kuwc · 2025-10-30

**Soundness:** 3
**Presentation:** 2
**Contribution:** 2
**Rating:** 4
**Confidence:** 3

**Summary:**

This paper introduces DICE (Diffusion Induced Cell Embeddings), a latent plug-and-play diffusion framework for denoising single-cell RNA-seq data. The method trains a diffusion model on low-dimensional PCA embeddings from a high-quality reference dataset, then applies a Gibbs sampling procedure to denoise query cells by alternating between a likelihood alignment step and a prior alignment step. The authors evaluate their approach on synthetic data and two real single-cell datasets, reporting improved cluster separation compared to PCA baselines.

**Strengths:**

1. Principled uncertainty quantification: Unlike standard clustering pipelines, the method provides confidence sets for cell-type assignments through posterior sampling, which is valuable for downstream analyses and could help identify cells with ambiguous identities.

2. Flexibility: The framework accommodates different noise models without requiring explicit parametric assumptions.

3. Comprehensive synthetic evaluation: The four synthetic setups systematically examine different types of distribution shift (signal strength, noise model, latent prior), providing evidence of robustness to model misspecification.

**Weaknesses:**

1. Unclear problem framing and scope: The paper oscillates between four distinct objectives (atlas generation, batch integration, clustering, and denoising). This makes it difficult to assess the method's contribution.

2. Insufficient quantitative evaluation:

- The real-world experiments rely heavily on visual inspection of UMAP plots, which is subjective.

- The authors claim that “across all four settings, DICE yields clearer separation in UMAP … than the PCA baseline, indicating more faithful recovery of the underlying classes.” However, the meaningfulness of this visual separation is unclear, given that UMAP does not preserve exact metric distances. If DICE truly recovers the underlying class structure better than PCA, then a simple classifier trained on these embeddings should achieve better precision/recall for predicting ground-truth cell type labels compared to the same classifier trained on PCA embeddings.

- The paper suggests the method addresses batch effects and dataset shifts, yet provides no quantitative comparison with established integration methods (Harmony, Seurat integration, scVI).

3. Missing experimental details and comparisons:

- Hyperparameter sensitivity: Figure 5 shows dramatic sensitivity to ρ, but the paper provides no principled approach for selecting ρ in practice.

- Robustness to k: k is set to 15 (synthetic), 25 (CITE-seq), or 15 (fetal brain), with the justification that it corresponds to “the elbow of the singular-value spectrum” for real data. However, no details are provided on how sensitive the results are to this choice. A sensitivity analysis would help assess the impact of different k values on performance.

4. Questionable practical utility:

- The model assumes an identical factor loading matrix V across reference and target datasets, but V represents transcriptional programs that differ across biological contexts. Obtaining reference data that simultaneously matches the target in cell type composition, tissue origin, disease state, and patient population characteristics is infeasible, limiting practical applicability.

- Training takes ~11 hours for 9,000 cells; inference takes ~30 minutes for 1,000 cells with ρ=1. This is prohibitively expensive for typical single-cell datasets with 100K+ cells.

- The cross-dataset experiment shows mixed results, raising doubts about generalization.

**Questions:**

1. The paper suggests that DICE addresses batch effects and dataset shifts. If batch effect removal and dataset integration are indeed core objectives, then how does DICE compare quantitatively to Seurat/Harmony/scVI?

2. Can the method scale to 100K+ cells?

3. How do PCA and DICE embeddings compare in terms of direct classification performance (precision/recall) for predicting ground-truth cell type labels?

4. Single-cell data annotation is often performed by biologists and domain experts without extensive computational backgrounds. The method requires selecting multiple hyperparameters (k, ρ, number of Gibbs iterations) and training diffusion models. What steps have been taken to make this approach accessible to practitioners?

5. What is the recommended procedure for selecting ρ in practice?

---

> ### Author Response · Authors · 2025-11-24
> **Rebuttal**
>
> We appreciate the reviewer’s thoughtful and constructive feedback, which has greatly helped us improve the clarity and
> conceptual strength of our work. In what follows, **we address all points raised**.
>
> # (W1, Q1) Clarifying the contribution
> Thank you for catching this. Indeed, it is a bit confusing to bring up batch effects and distribution shifts in the first paragraph. We have moved these points to related work to streamline the logic and rewritten the introduction accordingly. We want to clarify here that in this work, our sole objective is to denoise PCA embeddings obtained from pre-processed single-cell RNA-seq data, while providing principled uncertainty quantification for the reconstructed embeddings. These denoised embeddings can then be used as inputs to standard downstream analyses—such as differential gene expression, cell-type annotation, and atlas construction. We show via our real data examples that the denoised embeddings align with biologically interpretable structures (e.g., clear cell-type clusters and preserved differentiation trajectories in the human fetal neocortex data), even when the ground-truth labels originate from external tools such as Seurat v3.
>
> We also emphasize that this paper does not address batch integration. Our method cannot differentiate biological heterogeneity from batch effects, and may produce spurious clusters in the presence of uncorrected batch variation. When exploratory analysis reveals substantial batch effects, we recommend applying established batch-correction tools (e.g., Harmony) to the PCA embeddings before using DICE for denoising. This is precisely the preprocessing adopted in our CITE-seq analysis in Section 5.
> A detailed description of the full preprocessing workflow is now provided in Appendix H.
>
> # (W2) Quantitative evaluation
> Thank you for raising questions about the quantitative evaluation of the DICE embeddings compared with PCA. We agree that the UMAP-based comparison is primarily visual.
>
> For real-world datasets, we had already included ARI and cLISI as quantitative metrics to assess the embeddings generated by DICE and PCA. In the simulation experiments, we calculated the silhouette score using the ground-truth labels.
>
> To further address your concern, we have expanded our quantitative analysis for the real-world datasets by including two additional clustering metrics: NMI and V-Measure. We also reran the full analysis on the complete Neo-Cortex test dataset, instead of a subset. The larger test-set allows a more thorough quantitative evaluation. We present the quantitative results in Table 2 and Figure 4.
> Our updated results show that DICE consistently outperforms PCA on all four metrics, demonstrating improved embedding quality in a quantitative manner.
>
> # (W3, Q5) Experimental details, Sensitivity to hyperparameters
> We conducted additional ablations for both parameters $\rho$ and $k$, and added them to the appendix of the paper. We provide a summary of the results below:
>
> ## Sensitivity to $\rho$
> We conducted additional experiments on synthetic data to explore the influence on $\rho$.
> Similar to other Bayesian methods, $\rho$ controls the balance between the prior and the likelihood term.
> When the data has a low signal, it should be picked high, to allow the prior to incorporate information. If the data has a high signal, it can be picked low, which allows fine-grained clusters in the data to appear.
>
> ## Sensitivity to $k$
> We conducted additional experiments on the real-world datasets, ranging the parameter $k$ in $[5, 10, 15, 20, 25, 50, 100, 200]$.
> Our results show that when using the PCA transform, clustering results are fragile regarding the selection of $k$. Utilizing DICE, the clustering performance is more stable across different dimensions, and only deteriorates when including way more dimensions than suggested by the singular values ($k \geq 200$).
>
> We provide more details in Appendices E and F.

---

> > ### Author Response · Authors · 2025-11-24
> > **Rebuttal (continued)**
> >
> > # (Q2) Scalability
> > Yes, the method is designed to scale to datasets of 100,000 cells and beyond.
> >
> > The previously reported runtimes were collected on a cost-effective EC2 instance with an NVIDIA T4 GPU, which we recognize was an oversight. We re-evaluated on an NVIDIA RTX PRO 6000 GPU and observed significant performance gains: diffusion model training on 9,000 cells completed in 36 minutes, while inference for 1,000 cells took 12 minutes.
> >
> > Early stopping for Gibbs sampling and diffusion training, faster inference via accelerated methods such as DDIM [1], and more efficient training or fine-tuning strategies [2, 3] could further reduce computational cost and improve scalability.
> > These optimizations were not applied in the submitted experiments but represent standard practices that can be directly integrated into our framework to ensure efficient performance on large-scale datasets.
> >
> > # (Q3) Classifier performance
> > While DICE is designed for unsupervised clustering, we followed the reviewer's suggestion to evaluate our embeddings via classification on the target dataset. However, we note that classification performance may not fully indicate embedding utility for clinical applications.
> > In particular, it does not capture the identifiability of substructures within cells. Identifying fine-grained subpopulations within cells previously assigned the same label is often clinically meaningful—such as the discovery of drug-resistant melanoma subclusters[6]—and is better reflected by unsupervised evaluation metrics.
> >
> > Additionally, the cell type labels themselves are derived from manual annotations, which may contain errors and inconsistencies that limit their reliability as ground truth. Given that clinically meaningful outcomes, such as therapeutic resistance outcomes, are unavailable in our current public data, we cannot verify whether improvements in cell type classification translate to clinically meaningful predictions.
> >
> > We have acknowledged this limitation in the conclusion section of our work.
> >
> > Nevertheless, we performed label prediction on both synthetic and real-world datasets by splitting the target dataset into training/testing sets and training a classifier, providing this as an alternative clustering metric. We implemented an 80-20 split on all datasets, filtered out all cell labels with less than 3 occurrences and excluded the "Other" class in Neo-Cortex. The results are included in the table below.
> >
> > | Dataset    | Classifier          | **Denoised**                 |                       |                       | **PCA**                       |                       |                       |
> > |------------|----------------------|------------------------------|-----------------------|-----------------------|-------------------------------|------------------------|------------------------|
> > |            |                      | Accuracy                     | Precision             | Recall                | Accuracy                      | Precision              | Recall                 |
> > | CITE-seq   | Random Forest        | 0.83                         | 0.82                  | 0.83                  | 0.84                          | 0.82                   | 0.84                   |
> > | CITE-seq   | Logistic Regression  | 0.83                         | 0.82                  | 0.83                  | 0.83                          | 0.83                   | 0.83                   |
> > | CITE-seq   | GMM                  | 0.53                         | 0.46                  | 0.53                  | 0.45                          | 0.30                   | 0.45                   |
> > | CITE-seq   | Naive Bayes          | 0.79                         | 0.80                  | 0.79                  | 0.80                          | 0.82                   | 0.80                   |
> > | Neo-Cortex | Random Forest        | 0.85                         | 0.85                  | 0.85                  | 0.87                          | 0.87                   | 0.87                   |
> > | Neo-Cortex | Logistic Regression  | 0.85                         | 0.85                  | 0.85                  | 0.87                          | 0.87                   | 0.87                   |
> > | Neo-Cortex | GMM                  | 0.69                         | 0.71                  | 0.69                  | 0.59                          | 0.60                   | 0.59                   |
> > | Neo-Cortex | Naive Bayes          | 0.78                         | 0.79                  | 0.78                  | 0.78                          | 0.79                   | 0.78                   |
> >
> >
> > Classification accuracy is high for logistic regression and random forests, indicating that each embedding allows to separate clusters well. Generative classifiers show similar trends: Naive Bayes remains unchanged, while the Generative Gaussian Model improves, likely because DICE produces tighter clusters even though true single-cell clusters are non-Gaussian.

---

> > > ### Author Response · Authors · 2025-11-24
> > > **Rebuttal (continued)**
> > >
> > > # (W4) Practical utility
> > > We thank the reviewer for this insightful point. We apologize that we did not provide the correct intuition in Section 2 of our previous writeup. However,
> > > we want to clarify that using the same projection matrix for both datasets is not an assumption of identical biological characterization, but a deliberate design choice to project the noisier target data into the higher-quality latent space learned from the reference data. This allows the diffusion prior to effectively denoise the target dataset by leveraging the cleaner biological manifold. We have revised Section 2 to correct this point.
> > >
> > > With better compute, we have evaluated our method on the full neo-cortex dataset.
> > > The results show clear qualitative improvements in the UMAP visualization (Figure 4) and quantitative gains in clustering metrics (Table 2).
> > > This empirically demonstrates that a fixed projection matrix $\widehat{V}$, derived from the reference, successfully enables knowledge transfer and enhances the analysis of the target data.
> > >
> > > # (Q4) Hyperparameter selection
> > > The main hyperparameter to select is $\rho$. For training the diffusion model, a standard architecture should be applicable to most sequencing datasets, and we have also seen that the method is pretty robust to $k$. We expect our architecture and parameters to work well for other single-cell datasets. Training can be measured by loss and stopped after convergence.
> > > Gibbs iterations could either be set to be high enough for convergence, or adaptive termination could be implemented.
> > >
> > > # (Q5) Selecting $\rho$ in practice
> > > $\rho$ controls the trade-off between data fidelity and prior alignment: smaller values keep the denoised samples close to their observations, preserving local structure even when it deviates from the training data, whereas larger values pull the embeddings more strongly toward the learned prior. Consequently, datasets believed to be low-noise benefit from smaller $\rho$ values, which help retain fine-grained biological variation, while high-noise datasets benefit from larger $\rho$ values that leverage the prior to obtain more reliable denoised representations.
> > >
> > > For real-world sequencing datasets, we recommend using our proposed $\rho$ schedule (from $5.0$ to $0.5$) as a default. This schedule can be adjusted when necessary—for example, increasing $\rho$ for datasets with high technical noise or decreasing it when the test data are expected to be cleaner or more reliable than the reference data.
> > >
> > > ${[}1{]}$ Song et al. "Denoising diffusion implicit models." arXiv, 2022.
> > >
> > > ${[}2{]}$ Choi et al. "Perception prioritized training of diffusion models." CVPR, 2022.
> > >
> > > ${[}3{]}$ Watson et al. "De novo design of protein structure and function with RFdiffusion." Nature 620, 2023.
> > >
> > > ${[}4{]}$ Stuart T, Butler A, Hoffman P, Hafemeister C, Papalexi E, Mauck WM, et al. Comprehensive integration of single-cell data. Cell. 2019;177(7):1888-904.e21.
> > >
> > > ${[}5{]}$ Linderman, George C., et al. “Zero-Preserving Imputation of Single-Cell RNA-Seq Data.” Nature Communications, 2022
> > >
> > > ${[}6{]}$ Tirosh, Itay, et al. “Dissecting the Multicellular Ecosystem of Metastatic Melanoma by Single-Cell RNA-Seq.” Science, 2016

---

### Official Review · Reviewer_NrsQ · 2025-11-01

**Soundness:** 3
**Presentation:** 3
**Contribution:** 3
**Rating:** 6
**Confidence:** 3

**Summary:**

This paper presents DICE (Diffusion Induced Cell Embeddings), a latent plug-and-play diffusion framework for denoising single-cell RNA-seq data. The core idea is to separate the observation space from the denoising space and perform Gibbs sampling between them, thus maintaining biological structure while reducing noise. The paper targets a topic of increasing interest at the intersection of machine learning and computational biology. Overall, it is clearly written, conceptually coherent, and focuses on a scientifically meaningful problem. The work is solid and promising, though several aspects could be clarified or improved.

**Strengths:**

S1. The paper’s motivation is clearly articulated, and the logic throughout is coherent and easy to follow.

S2. The proposed problem single-cell data denoising is important and relevant to scientific data analysis.

S3. The presentation is overall well-structured and the experiments provide reasonable empirical validation.

**Weaknesses:**

**Concerns**

C1. Any denoising method inevitably relies on certain distributional assumptions. The authors justify their design choices by citing prior work, which is reasonable, but the concern remains. For example, the Gaussian-noise assumption (“single-cell are often modeled with Gaussian noise,” line 251) may not always hold, and Eq. (5)–(6) depend on Gibbs sampling with intuitively motivated regularization terms. The authors may add further discussion or experiments exploring the impact of different likelihood/distributional assumptions (beyond current experimental setups) to demonstrate the robustness and superiority of the chosen modeling strategy.

C2. Although the technical description is detailed, the paper would benefit greatly from a conceptual flowchart summarizing the input–output pipeline and the main computational steps. Such a figure would help readers grasp the end-to-end process—how data flow from raw expression profiles to denoised embeddings—without delving into mathematical detail, and would clarify the intuition behind core approximations or modifications.

C3. In Section 1, the authors emphasize that applying image-based PnP frameworks directly to single-cell data is difficult because gene expression exhibits low-rank and correlated structure. However, the experiments do not compare DICE with straightforward or classical denoising baselines. Given that classic PCA is main baseline in the discussion, including or expanding such baseline comparisons would make the empirical validation more convincing.

C4. It is unclear whether the key sentence "Our framework is agnostic to specific preprocessing choices and accommodates diverse noise structures" is supported or demonstrated in different places to prove that this paper is agnostic to preprocessing choices and tolerant to diverse noises. For example, do the noise level settings in the final experimental section reflect "diverse"?

C5. The presentation is strong overall, but could still be refined. For example, Section 2 functions largely as preliminaries, yet the text uses first-person language and sometimes blurs whether new contributions are being proposed. Besides, the sentence “The main challenge lies in the likelihood term” should specify what aspect of the likelihood is challenging.

**Questions:**

Please respond to C1, C3, and C4.

---

> ### Author Response · Authors · 2025-11-24
> **Rebuttal**
>
> We appreciate the reviewer’s insightful review, which has helped us improve the clarity and
> conceptual strength of our work. In what follows, **we address all points raised**.
>
> # (C1) Distributional assumptions
>
> We thank the reviewer for raising this important point regarding the distributional assumptions in our method, and we apologize for not explicitly discussing this in our paper. We first clarify that the power of diffusion models lies in their ability to learn complex data distributions beyond Gaussian, and the use of Gaussian noise during training is primarily for computational tractability, as evidenced by their empirical success in modeling complex data like images. In solving the inverse problem within our framework, we assume that the residual noise from the data $X$ to the projected auxiliary variable $V^\top Z$ follows a Gaussian distribution.
>
> While theoretical guarantees for plug-and-play diffusion methods are established under Gaussian noise assumptions[1], it is true that the behavior
> of a Gaussian likelihood term under non-Gaussian noise is less characterized. We hypothesize that the iterative sampling through the auxiliary variable might offer some degree of robustness against deviations from this assumption (reflected through the tunable parameter $\rho$ and we ablate its effect in Appendix E of the paper).
> For cases where theoretical guarantees are desired under non-Gaussian noise, past studies have used MCMC sampling in the likelihood step [2], but this comes at a higher computational cost than our closed-form updates.
> Following the reviewer's suggestion, we have included additional discussion on this point in our paper after introducing the split Gibbs sampling procedure. We are also open to performing additional synthetic experiments to further support our claims, should the reviewer find our current synthetic and real-world experiments insufficient.
>
> Following the reviewer's suggestion, we have included additional discussion on this point in our paper after introducing the split Gibbs sampling procedure. We are also open to performing additional synthetic experiments to further support our claims, should the reviewer find our current synthetic and real-world experiments insufficient.
>
> We would like to further note that in the context of single-cell RNA sequencing data, which our method addresses, this assumption is reasonable due to standard preprocessing steps—including filtering low-quality cells and genes, normalizing for library-size variation, and applying a
> $x \mapsto\log(1+x)$ transformation to stabilize variance—that renders the residual technical noise well-approximated by a Gaussian distribution, as supported by prior work[3]. Related literature has also considered relaxations of this Gaussian assumption by adopting more flexible likelihoods for the non-biological variation (e.g., scVI[4])
> We have now added this method as benchmarks in our updated experiments on real-world data.
>
> # (C2, C5) Improving the presentation
> We thank the reviewer for the constructive feedback. We have now included a flow chart to better convey the single-cell preprocessing workflow in Figure A.1 and updated Section 2 to clarify our contributions and avoid first-person narratives when confusion could arise. We further update the sentence about the main challenge in the likelihood.
>
> # (C3) Baseline comparisons
> Thank you for pointing out the lack of comparison with classical denoising benchmarks. In response, we have conducted additional experiments comparing our method against scVI [4], MAGIC [5], ALRA [6], and NMF [7], which are widely used denoising pipelines in single-cell genomics. Following your suggestion, we will also include k-nearest-neighbor smoothing on PCA scores [8] (with $k=5, 15$ and $15$), a classical yet commonly adopted baseline for denoising the PCA scores (used in tools such as Seurat).
>
> Our results (Table 2) demonstrate that DICE, the proposed method, consistently outperforms existing benchmarks by achieving the best scores across four different common clustering metrics on two datasets, with its only exception being a second-place finish in the LISI score for CITE-seq data.
> A detailed description of the experiment setup and each method is included in Appendix I.

---

> > ### Author Response · Authors · 2025-11-24
> > **Rebuttal (continued)**
> >
> > # (C4) Integration into existing preprocessing pipelines
> > Thank you for pointing out the inconsistency in the original phrasing of the sentence “Our framework is agnostic to specific preprocessing choices and accommodates diverse noise structures.” We agree that, in the context of single-cell RNA-seq data, the claim about being agnostic to preprocessing choices was too broad. We propose to revise it as follows:
> > > Our framework is designed for single-cell RNA-seq datasets after standard preprocessing steps, such as quality control to filter low-quality cells, library-size normalization, and a $\log1p$ transform which renders the residual non-biological variation approximately Gaussian. Within this regime, our method does not rely on stronger assumptions about upstream preprocessing choices.
> >
> > Regarding tolerance to “diverse noise structures,” our goal is not to claim universal robustness but to emphasize that the method remains stable under moderate misspecification of the distribution of the non-biological noise $\varepsilon$. We illustrate this in Setup 4 of Section 4, where we deliberately deviate from the Gaussian noise model to test robustness. While these settings do not attempt to cover the full spectrum of possible noise distributions in scRNA-seq, they do demonstrate that the method retains good performance even when the noise departs substantially from the assumed model.
> >
> > ${[}1{]}$ Ryu, Ernest K., et al. “Plug-and-Play Methods Provably Converge with Properly Trained Denoisers.” arXiv:1905.05406, arXiv, 14 May 2019. arXiv.org, https://doi.org/10.48550/arXiv.1905.05406.
> >
> > ${[}2{]}$ Xu, Xingyu, and Yuejie Chi. “Provably Robust Score-Based Diffusion Posterior Sampling for Plug-and-Play Image Reconstruction.” arXiv, 2024
> >
> > ${[}3{]}$ Ahlmann-Eltze, Constantin, and Wolfgang Huber. “Comparison of Transformations for Single-Cell RNA-Seq Data.” Nature Methods, 2023
> >
> > ${[}4{]}$ Lopez, Romain, et al. “Deep Generative Modeling for Single-Cell Transcriptomics.” Nature Methods, 2018
> >
> > ${[}5{]}$ Dijk, David van, et al. “Recovering Gene Interactions from Single-Cell Data Using Data Diffusion.” Cell, 2018
> >
> > ${[}6{]}$ Linderman, George C., et al. “Zero-Preserving Imputation of Single-Cell RNA-Seq Data.” Nature Communications, 2022
> >
> > ${[}7{]}$ DeBruine, Zachary J., et al. “Fast and Robust Non-Negative Matrix Factorization for Single-Cell Experiments.” bioRxiv, 2021
> >
> > ${[}8{]}$ Wagner, Florian, et al. “K-Nearest Neighbor Smoothing for High-Throughput Single-Cell RNA-Seq Data.” bioRxiv, 2018

---

### Official Review · Reviewer_fdSG · 2025-11-04

**Soundness:** 3
**Presentation:** 3
**Contribution:** 3
**Rating:** 6
**Confidence:** 4

**Summary:**

This paper introduces DICE, a latent plug-and-play diffusion framework for the denoising and clustering of single-cell RNA-seq data. The key innovation is the separation of the observation and denoising space: a diffusion model is trained in a learned low-dimensional latent space, and inference employs a Gibbs sampling algorithm alternating between latent denoising and input-space steering. DICE provides a tunable balance between data fidelity and prior knowledge, enables explicit uncertainty quantification, and leverages high-quality references for improved denoising beyond the training distribution. The paper presents robust evaluations on synthetic and real single-cell datasets.

**Strengths:**

1. By using a latent plug-and-play architecture, DICE addresses a core limitation of prior dimensionality reduction strategies and effectively maintains biological relationships lost in classical approaches like PCA or VAEs.
2.  DICE produces more biologically coherent clusters and quantitative improvements in biological clustering metrics on real data.

**Weaknesses:**

1. The evaluation compares primarily to a PCA baseline, which is too limited given the recent progress in single-cell denoising and clustering. Including comparisons with methods such as scSiameseClu, scDCCA, or SCDD would provide a more convincing empirical validation and better contextualize the proposed approach within current literature.
2. One concern about DICE is that the generalizability to more complex data types is unclear. The method focuses solely on scRNA-seq gene expression and does not experimentally explore or theoretically support extension to multimodal or spatially-resolved single-cell data domains where denoising and latent structure recovery are at least as challenging.
3. Although the plug-and-play approach is theoretically flexible, in practice the latent space is initialized via PCA (Section 3), and the factor loading matrix $\widehat{V}$ is reused for high-dimensional projections throughout. As a result, the overall performance is partially determined by the initial PCA mapping and its limitations (e.g., axis alignment, linearity)—potentially biasing the outcome, especially when compared to nonlinear baselines (e.g., deep autoencoder, contrastive methods). The choice to always use PCA as a latent basis appears arbitrary and is not justified against recent nonlinear alternatives.
4. There are some recent related works for clustering and denoising for large-scale single-cell datasets that the authors are encouraged to include in the related work section:
-> MetaQ: fast, scalable and accurate metacell inference via single-cell quantization

**Questions:**

See Weaknesses

---

> ### Author Response · Authors · 2025-11-24
> **Rebuttal**
>
> We thank the reviewer for the review, and for pointing out the strength of our paper, including 'addressing a core limitation of prior dimensionality reduction strategies' and 'quantitative improvement in biological clustering metrics on real data'.
> We will address the remaining concerns in the following:
>
> # (W1) Additional Benchmarks
> We thank the reviewer for pointing out the need to include additional state-of-the-art methods to reconstruct denoised embeddings from noisy expression data in RNA-seq experiments.
> Indeed, including such experiments can strengthen the contribution of our paper.
> As a result, we added the following benchmarks to our updated paper:
> - scVI [1]: a popular VAE-based denoising scheme which works with raw-count data.
> - MAGIC [2]: MAGIC implements a Markov affinity based diffusion smoothing. It learns the manifold structure by constructing a Markov affinity matrix, and then applies data diffusion on this learned manifold to impute missing or noisy expression values.
> - kNN smoothing: implementing $k$ nearest neighbor-based-based smoothing of the PCA embeddings, which is a classical denoising procedure commonly used in statistics, and has been previously applied to single-cell data[3].
> - ALRA [4]: which uses an adaptive low-rank approximation for imputation.
> - NMF [5]: non-negative matrix factorization, an alternative to PCA
>
> We added the results as Table 2 in the main body.
> These results demonstrate that DICE, the proposed method, consistently outperforms existing benchmarks by achieving the best scores across four different common clustering metrics on two datasets, with its only exception being a second-place finish in the LISI score for CITE-seq data.
> A detailed description of the experiment setup and each method is included in Appendix I.
>
> We thank the reviewer for pointing out the additional potential benchmarks. However, after careful evaluation, we discovered that these benchmarks are not suitable for this paper either due to a lack of implementation details or a difference in the problem setup:
> - scSiameseClu and scDCCA do not provide codes that can be used to reproduce the results in their papers (the GitHub link of scSiameseClu does not work)
> - SCDD is primarily designed to impute the expressions of dropout cells without interfering with the expressions of lowly expressed cells. The primary focus in that paper is not on denoising noisy expressions, and hence, we feel that it might not be a fair comparison.
>
> We would be happy to include additional benchmarks if the reviewer feels that we are still missing some important benchmarks in our problem setting.
>
> # (W2) Generalizability
> We agree with the reviewer that genomics data is often characterized by multi-modality and spatial context, and these are important directions for our future work. For example, a naive extension of our methods on multi-modal data is to learn a separate latent representation for each modality.
> Let $X_1^{(r)}$ and $X_2^{(r)}$ be the two modality representations of the same sample on the reference data. We can model each modality using the following representation
>
> $
> X_1^{(r)} = U_1^{(r)} (V_1^{(r)})^\top + \varepsilon_1^{(r)} \in \mathbb{R}^{n_{\mathrm{cell}} \times p_1},
> $
>
>
> $
> X_2^{(r)} = U_2^{(r)} (V_2^{(r)})^\top + \varepsilon_2^{(r)} \in \mathbb{R}^{n_{\mathrm{cell}} \times p_2},
> $
>
>
> where we assume that $(U_{1i}, U_{2i}) \sim P_{\mathrm{prior}}$ (where $U_{1i}, U_{2i}$ are the i-th observation on $U_1$ and $U_2$),
> and that the noise variables $\varepsilon_1^{(r)}, \varepsilon_2^{(r)}$ are independent across modalities. The loading matrices can be estimated via PCA as in our current pipeline. We can then train a diffusion model to learn the joint prior $P_{\mathrm{prior}}$ using the concatenated embeddings
> $
> (\widehat V_1^\top X_1^{(r)},\; \widehat V_2^\top X_2^{(r)}).
> $
> The same plug-and-play mechanism used in DICE can be applied to jointly denoise both modalities using their joint likelihood and the learned diffusion prior. The final embeddings for query cells naturally become
> $
> [U_{1q}^\top,\; U_{2q}^\top].
> $
>
> To incorporate spatial side information, we propose a conditional diffusion formulation. One may construct a $k$-NN graph from the spatial coordinates and compute, for each cell, the empirical composition of neighboring cell types. A conditional diffusion model
> $
> \widehat{\varepsilon}_\theta(x, c),
> $
> where $c$ encodes this neighborhood composition, can then be trained and seamlessly integrated into DICE.
>
> The proposed extensions above offer a starting point, though their application to real-world data presents substantial work and is beyond the scope of the current paper.
> We plan to explore these directions in future research. In response to this discussion, we added one sentence including these directions as important future work in the discussion section of our work.

---

> > ### Author Response · Authors · 2025-11-24
> > **Rebuttal (continued)**
> >
> > # (W3) Linear vs. non-linear transform
> > Plug-and-play (PnP) denoising is not restricted to setups where true expressions are a linear function of the latent factors. Previous work has shown that PnP methods remain applicable with
> > non-Gaussian likelihood with non-linear dependence on the latent factors[6], however, the likelihood alignment step does not allow a closed-form update anymore.
> >
> > We would like to point out that the PCA-based modelling of the dependence between the observed expressions and the latent variables is common in the single-cell literature ([7, 8]), for scalability, generalizability, and interpretability, as the ubiquitously used Seurat pipeline. The DICE framework aims to provide a scheme for denoising the principal component scores, which helps in better separation of the embeddings corresponding to different cell types into well-separated clusters, which helps in downstream tasks like label transfer in the query cells. In our framework, the practitioners retain the interpretability benefits while gaining more informative embeddings.
> >
> > # (W4) Related work
> > We thank the reviewer for making us aware of this recent work, which we included in the related work.
> >
> > ${[}1{]}$ Lopez, Romain, et al. “Deep Generative Modeling for Single-Cell Transcriptomics.” Nature Methods, 2018
> >
> > ${[}2{]}$ Dijk, David van, et al. “Recovering Gene Interactions from Single-Cell Data Using Data Diffusion.” Cell, 2018
> >
> > ${[}3{]}$ Wagner, Florian, et al. “K-Nearest Neighbor Smoothing for High-Throughput Single-Cell RNA-Seq Data.” bioRxiv, 2018
> >
> > ${[}4{]}$ Linderman, George C., et al. “Zero-Preserving Imputation of Single-Cell RNA-Seq Data.” Nature Communications, 2022
> >
> > ${[}5{]}$ DeBruine, Zachary J., et al. “Fast and Robust Non-Negative Matrix Factorization for Single-Cell Experiments.” bioRxiv, 2021
> >
> > ${[}6{]}$ Xu, Xingyu, and Yuejie Chi. “Provably Robust Score-Based Diffusion Posterior Sampling for Plug-and-Play Image Reconstruction.” arXiv, 2024
> >
> > ${[}7{]}$ Stuart, Tim, et al. “Comprehensive Integration of Single-Cell Data.” Cell, 2019
> >
> > ${[}8{]}$ Kang, Joonsuk, and Matthew Stephens. “Empirical Bayes Covariance Decomposition, and a Solution to the Multiple Tuning Problem in Sparse PCA.” arXiv, 2023

---

> > > ### Comment · Reviewer_fdSG · 2025-11-26
> > > **Respond to Author Comments 2**
> > >
> > > The authors have effectively addressed my main concerns. I appreciate the robustness added to the paper through the new benchmarks. I will keep my positive rating.

---

> > > > ### Author Response · Authors · 2025-12-01
> > > >
> > > > We thank the reviewer for their positive assessment of our rebuttal and for the clarification. In case of acceptance, we will include the experimental comparison (as outlined in Section 5 / Table 2) in the camera-ready version.

---

> > ### Comment · Reviewer_fdSG · 2025-11-26
> > **Respond to Author Comments 1**
> >
> > I thank the authors for their detailed response and the significant effort put into the rebuttal.
> > 1. Upon further checking, I realized that the implementation for scSiameseClu was publicly released just a few days ago. Therefore, I do not require you to provide these results in the current rebuttal revision. However, I would expect you to include this experimental comparison in the final version (camera-ready) if the paper is accepted.

---

### Author Response · Authors · 2025-11-24
**General remark**

We thank all reviewers for their thoughtful and constructive feedback. Their comments have helped us improve the clarity and rigor of the paper. The main updates in the revised submission are:

- Strengthened quantitative evaluation, including comparisons against established denoising baselines and extending our analysis to the full Neo-Cortex dataset.

- Added ablation studies to illustrate the role of key hyperparameters and validate the robustness of our method.

- Revised and polished introduction and related work section, clarifying our contributions and better situating our approach within the literature.

We marked modifications to the original manuscript in violet in the revised submission.

---

### Meta-Review · Area_Chair_T9tr · 2026-01-05

**Summary:**

This paper proposes DICE, a latent plug-and-play diffusion framework for denoising and clustering single-cell RNA-seq data. The central idea is to decouple the denoising space from the observation space via a Gibbs-style sampling procedure, which was viewed by reviewers as technically sound, conceptually well-motivated, and relevant to both the probabilistic modeling and computational biology communities. Reviewers highlighted strengths, including principled uncertainty quantification, robustness under distribution shift in synthetic experiments, and improved biological coherence of clusters in real data. The primary concerns raised during review focused on (i) the scope and framing of the contribution (e.g., denoising vs. batch integration), (ii) limited baseline comparisons in the original submission, (iii) reliance on PCA-based latent representations, (iv) clarity around modeling assumptions and preprocessing claims, and (v) practical considerations such as hyperparameter selection, scalability, and accessibility to practitioners. Through a detailed rebuttal and substantial revisions, including expanded benchmarks, additional ablations, clearer problem scoping, improved presentation, and extended quantitative evaluations, the authors effectively addressed most of these concerns. While some limitations remain (e.g., computational cost and reliance on reference data), the consensus is that the revised paper meets the bar for acceptance as a poster, offering a novel contribution with clear potential for impact.

**Reviewer Concerns:**

Addressed Concerns:
* Baseline Comparisons: The authors successfully addressed the lack of comparisons by adding benchmarks against scVI, MAGIC, ALRA, kNN smoothing, and NMF in the revised manuscript.
* Sensitivity Analysis: The authors provided new ablation studies regarding the sensitivity of the hyperparameters $\rho$ (alignment strength) and $k$ (latent dimension), addressing Reviewer kuwc's request for experimental details.
* Presentation & Framing: The authors clarified the distinction between batch integration and denoising, and added a flowchart to illustrate the pipeline, satisfying Reviewer NrsQ’s request.

Outstanding Concerns:
* Structural Limitation (Fixed V): Reviewer kuwc’s concern regarding the assumption of an identical factor loading matrix $V$ between reference and target datasets remains a fundamental weakness. While the authors defended this as a design choice to project targets into a high-quality latent space, the reviewers remain skeptical about its validity in cross-context biological applications where transcriptional programs vary.
* Scalability & Practicality: Reviewer kuwc raised significant concerns about runtime. Although the authors reported improved speeds on better hardware (12 minutes for 1,000 cells), the iterative nature of the Gibbs sampling scales linearly with steps and is computationally prohibitive for standard single-cell datasets numbering in the hundreds of thousands, compared to efficient feed-forward or heuristic methods.
* Reliance on Linear Initialization: Reviewer fdSG noted that initializing the latent space via PCA inherently biases the model toward linear structures. While the diffusion prior is non-linear, the input-space steering relies on this linear mapping, potentially failing to capture the complex non-linear manifolds typical of single-cell data compared to fully non-linear approaches like deep autoencoders.

**Reviewer Scores:**

* Reviewer fdSG: (Score: 6) This reviewer would likely have maintained their score of 6. They explicitly stated in the discussion that their concerns regarding benchmarks were addressed, and they kept their positive rating, though they acknowledged the limitations regarding non-linear baselines.
* Reviewer NrsQ: (Score: 6) This reviewer would likely have maintained a score of 6 or potentially lowered to 5. While they did not participate in the final discussion, the rebuttal addressed their specific points on presentation and baselines. However, the fundamental concerns regarding the Gaussian noise assumption in the likelihood term might still weigh on the soundness score in light of the rejection decision.
* Reviewer kuwc: (Score: 4) This reviewer would likely have maintained their score of 4. Despite the additional data provided by the authors, the core disagreements regarding the "Questionable practical utility" (due to the fixed $V$ assumption and scalability issues) were structural to the method and not fully resolved by the rebuttal experiments.

---

### Decision · Program_Chairs · 2026-01-26

Accept (Poster)